# Mother Schema, Obstetric Dilemma, and the Origin of Behavioral Modernity

**DOI:** 10.3390/bs9120142

**Published:** 2019-12-06

**Authors:** Richard Parncutt

**Affiliations:** Centre for Systematic Musicology, University of Graz, 8010 Graz, Austria; parncutt@uni-graz.at

**Keywords:** behavioral modernity, evolutionary psychology, mother schema, obstetric dilemma, language, music, origin, religion, reflective consciousness

## Abstract

What triggered the emergence of uniquely human behaviors (language, religion, music) some 100,000 years ago? A non-circular, speculative theory based on the mother-infant relationship is presented. Infant “cuteness” evokes the infant schema and motivates nurturing; the analogous mother schema (MS) is a multimodal representation of the carer from the fetal/infant perspective, motivating fearless trust. Prenatal MS organizes auditory, proprioceptive, and biochemical stimuli (voice, heartbeat, footsteps, digestion, body movements, biochemicals) that depend on maternal physical/emotional state. In human evolution, bipedalism and encephalization led to earlier births and more fragile infants. Cognitively more advanced infants survived by better communicating with and motivating (manipulating) mothers and carers. The ability to link arbitrary sound patterns to complex meanings improved (proto-language). Later in life, MS and associated emotions were triggered in ritual settings by repetitive sounds and movements (early song, chant, rhythm, dance), subdued light, dull auditory timbre, psychoactive substances, unusual tastes/smells and postures, and/or a feeling of enclosure. Operant conditioning can explain why such actions were repeated. Reflective consciousness emerged as infant-mother dyads playfully explored intentionality (theory of mind, agent detection) and carers predicted and prevented fatal infant accidents (mental time travel). The theory is consistent with cross-cultural commonalities in altered states (out-of-body, possessing, floating, fusing), spiritual beings (large, moving, powerful, emotional, wise, loving), and reports of strong musical experiences and divine encounters. Evidence is circumstantial and cumulative; falsification is problematic.

## 1. Introduction

Archeological evidence points to a gradual but profound change in human behavior between roughly 200,000 and 60,000 years ago, toward the end of the Middle Paleolithic in Europe and Middle Stone Age in Africa [1,2,3]. During this time of changing climate, *Homo sapiens* (anatomically modern humans) encountered and gradually displaced the Neanderthals. The behavioral change has been variously called “human revolution” [4], symbolic revolution” [5], “creative explosion” [2], “great leap forward” [6], or “Upper Palaeolithic revolution” [7]. The result has been labelled “behavioral modernity” [8]. 

The transition may have involved an anatomical or physiological development [9] that promoted cognitive processing of language [1], rapid speech [10], or timbral distinctions [11]. But given the gradualness and evolutionary recency of the change, it may also have been fundamentally cultural in nature [12,13]. Biological and cultural evolution interact [14]. 

The evidence for symbolic behaviors [15] and proto-art in the archaeological record is indirect. The evidence includes lithics (stone tools), artefacts (bone, antler), marks on stone surfaces (“cave art”), grave goods, body decorations (beads, pendants), pigments (ochre), functional spatial organization in habitations, and underground food storage. There is also evidence for long-distance trade, migration to colder northern latitudes, and sea travel to colonize other continents [7]. Taken together, these phenomena suggest “emergence of self-awareness and group identity, social diversification, formation of long-distance alliances, the ability to symbolically record information” [7] (p. 369). But the most interesting developments were archeologically largely invisible: language, religion, music, and reflective consciousness.

Theoretical physicists have searched in vain for a “theory of everything” that brings together general relativity and quantum field theory under an elegant mathematical roof [16]. The prehistoric emergence of behavioral modernity is another of science’s great mysteries. At first glance, it might be explicable simply by selection for higher intelligence. But behavioral modernity also involves universal, quasi-irrational behaviors such as religion and music. While these could involve taking elements of emotion and cognition out of the context for which they evolved (for example, infant attachment) and “intelligently” applying them elsewhere, beliefs in anthropomorphized supernatural agents with whom humans communicate cognitively and emotionally contradict accepted elements of intelligence such as logic, reasoning, critical thinking, and problem solving. The universality of religious beliefs and musical rituals involving transcendental experiences suggest the existence of an additional factor.

### 1.1. Evolution of the Mother-Infant Relationship

Inspired by Falk [17] and Dissanayake [18,19], I will present a psychological theory of the origin of behavioral modernity that is based on the mother-infant relationship and its evolution. The theory is intended to be parsimonious (conceptually simple, internally consistent, focusing on main effects, neglecting smaller issues), generalizable (explaining a wide range of observations), and fecund (generating new hypotheses and predictions). Specifically, I will investigate the possibility that the *obstetric dilemma* was the ultimate trigger that enabled and motivated behavioral modernity. This overarching thesis incorporates several interacting elements:

*Obstetric dilemma.* As early humans became increasingly bipedal, the shape and orientation of the pelvis and birth canal changed. In addition, brain size increased. Both changes exacerbated the birth process: infants were born earlier (relative to when they would otherwise have been born) and therefore became more fragile [20,21]. The evolutionary fitness of human infants depended increasingly on the ability of both mothers and infants to contribute actively to new, more complex forms of communication (e.g., motherese).

*Language.* Infants were more likely to survive if the foundations of proto-language—including a large vocabulary of learned relationships between complex abstract meanings and arbitrary sound patterns—were laid before birth. The word “learn” is used here in the usual psychological sense. It applies to human and non-human animals equally. Learning means acquiring new information, habits, or abilities; any contact with the environment that causes a lasting behavioral change (physical responses; perceptual/cognitive changes) may be regarded as learning [22]. Phenotypic variation in the cognitive abilities of newborns (including the ability to imitate complex sound patterns [11]) meant that the more cognitively able were more likely to survive to reproductive age. Whereas this principle presumably applies to all mammals, the obstetric dilemma in early humans meant that infant survival depended increasingly directly on the active contribution of infants to maternal-infant bonding. The resultant improvement in fetal cognitive abilities was an evolutionary adaptation—a response to new conditions imposed by the obstetric dilemma—that increased the resilience of human infants. 

*Mother schema.* Mother schema (MS) is presumably the first and most important psychological schema in ontogeny, organizing stimulus patterns attributable to the mother and her physical/emotional states from the perspective of the fetus or infant. Prenatal MS includes the sound of the mother’s voice, heartbeat, footsteps, and digestion, the feeling of her body movements, and associated biochemical changes (e.g. cortisol, drugs, neurotransmitters)—all from the fetal perspective. Postnatal MS involves different stimulus patterns. The early human fetus/infant adapted to the new situation created by the obstetric dilemma (the infant’s unprecedented fragility) by assigning extra cognitive resources to pre- and postnatal MS, strengthening mother-infant attachment.

*Operant conditioning.* Animal behavior can be conditioned by rewards and punishments. As MS became more salient and complex, it was more often activated later in life (during childhood and adulthood). In the context of early human ritual, repetitive stimulus patterns similar to those perceived before birth or during infancy evoked MS and associated emotions. These mysterious feelings (and corresponding biochemicals such as opioids and neuropeptides) acted as rewards (positive reinforcements), motivating ritual participants to repeat the actions that appeared to have caused the feelings. In proto-religious rituals, MS was evoked by combinations of subdued light, low-frequency acoustic resonances, unusual smells or tastes, unusual body postures, and/or changed states of consciousness brought about by diverse means, such that participants sensed the presence of mysterious supernatural agents. For music, early forms of singing, drumming, or dance evoked MS if they were perceived as similar to maternal voice, heartbeat/footsteps, or body movements during walking from the fetal perspective. 

*Religion.* MS (or a generalized ontogenetic transformation of MS) was activated in children and adults in ritual-like situations that were somehow similar to the prenatal world. When MS was evoked, special, strange, non-everyday emotions were experienced; these may be considered psychologically real even in the absence of conscious reflection if they correspond to specific neurophysiological patterns or mechanisms. In the absence of prenatal/infant episodic memory [23], the emotions evoked by early rituals were inexplicable for early humans—just as the origin of the universe or the disappearance of the soul after death (common-sense dualism [24]) was inexplicable. The experiences were similar to those one might have in the presence of a god-like being (large, moving, wise, loving, and/or powerful). Searching for causal explanations, early humans developed diverse religious narratives and beliefs (proto-religion). The feelings of caring and attachment that emerged when MS was evoked contributed to the development of religious power/dominance hierarchies. These parallel ontogenetic-psychological paths—from MS to adult attachment and from powerful adults to dominance hierarchies—may have reinforced the psychological foundations of religious experience and other aspects of human sociality, strengthening existing dominance hierarchies that were previously based on aggression alone.

*Music.* As early humans repeated the behaviors that produced these special experiences, proto-music (proto-melody, proto-rhythm, proto-dance) emerged. What we today recognize as “music” or “religion” developed more slowly, in conjunction with reflective consciousness and cultural development.

*Reflective consciousness.* Human reflective consciousness, also called self-reflection or self-reflective consciousness (cf. [25]) and defined here as the overt ability to observe one’s own consciousness and exercise introspection, also emerged gradually, in parallel with language, religion, and music. Pre- or corequisites for this development included theory of mind and mental time-travel (e.g., autobiographical memory), although different brain regions were involved [26].

### 1.2. Structure of the Thesis

Figure 1 is a preliminary attempt to illustrate possible causal relationships between and among the elements of the thesis. A simple figure of this kind can hardly do justice to the complexity of the processes under consideration. The figure therefore focuses on processes that are assumed most important or interesting for the present purpose.

The arrows illustrate hypothetical cause-effect relationships. The main arrows point downward. They reflect the following assumptions: plausible arguments for origins of human behaviors must be non-circular, linking them causally to pre-human phenomena; the main behavioral phenomena of interest here (religion, music, language, reflective consciousness) might not exist without the causal relationships illustrated by the downward arrows. Of the four main phenomena under consideration, reflective consciousness has a special status, because it is not directly observable [27]; experts are therefore more likely to disagree about the processes that enabled its emergence (the lower right part of the diagram).

The label “More complex MS” in the figure alludes to an intergenerational process whereby increasing infant fragility caused the fetus/infant to assign more cognitive resources to processing information about the mother and her changing physical and emotional states. Infants that were more skilled in this aspect of bonding were more likely to interact successfully with older humans (especially, but not only, the mother/carer) and hence to survive to reproductive age.

The figure illustrates how operant conditioning can account for the earliest proto-religion and proto-music. The reward or reinforcement that conditioned such behaviors was the special emotion (not necessarily positive, but mysterious) experienced when MS was activated in children or adults. 

### 1.3. “Hard Problems”

Chalmers [28] identified the “hard problem of consciousness”: How can we explain human subjective experience and solve the mind-body problem? Why do sentient organisms have *qualia* (phenomenal experiences)? More empirically accessible questions include: Why is human reflective consciousness (when analyzed developmentally [29]) so complex by comparison to consciousness in non-human animals?

Consciousness is sometimes explained by recursion. Referring to the level-of-consciousness model of Zelazo [29], Morin [30] explained that “the contents of consciousness are fed back into consciousness so that they can become accessible to consciousness at a higher level” (p. 364). Regardless of the truth content of this statement, its inherent circularity is logically problematic. *Circular reasoning* (related to *begging the question* and *infinite regress*) is a commonly encountered logical error [31].

Rowlands [32] pointed out that higher-order-thought models of consciousness, according to which a mental state is conscious only if the subject is thinking about that state, are generally circular. Rosenthal [33] clarified:
An explanation of what it is for mental states to be conscious either will itself appeal to mental states or it will not. Suppose now that all mental states are conscious, and that our explanation of what it is for mental states to be conscious does invoke mental states. Such an explanation will be circular, since the appeal to mental states is then automatically an appeal to conscious states. Invoking the very phenomenon we want to explain trivializes the explanation and prevents it from being informative.(p. 735)

Circular reasoning can be avoided by basing a theory of the origin of consciousness on pre-human behaviors and situations. Examples include anatomical changes in the vocal apparatus associated with upright walking [34]. The same applies to other aspects of behavioral modernity, each of which has its own “hard” (or enduringly intractable) problem:

*Language.* How and why did human communication become so enormously complex in its arbitrary sound patterns, extensive vocabularies, and hierarchical structures—relative to the poor linguistic abilities of non-human primates? The literature addressing this question is old, extensive, diverse, and unresolved. Whereas a descended larynx may initially have had nothing to do with language [35], postnatal laryngeal descent [36] and increased laryngeal control [37] clearly played important roles in language evolution, increasing the number of distinguishable sounds [38]. Brain enlargement may have led to laryngeal descent [39] and vice-versa [40,41]: the benefits of more complex communication may have meant selection for encephalization [42]. What, apart from bipedalism and encephalization, ultimately triggered or drove these complex interacting changes?

*Religion.* Why have all known human cultural groups produced belief systems that involve supernatural agents (human/animal gods/spirits) that created the universe and with whom humans can communicate personally? A clear answer should distinguish between social/emotional and cognitive/philosophical aspects. It should explain why people at all times and in all places have participated in religious rituals that do not contribute directly to survival and reproduction, but are highly costly, diverting time and energy from existential activities of finding and preparing food, caring for infants, and defending against attack. 

*Music.* How can extraordinary, powerful, transcendental, life-changing musical experiences be explained? Gabrielsson [43] asked almost a thousand respondents to describe the strongest experience with music they had ever had. No currently accepted music-psychological theory can account for his rich data. A successful theory should in addition explain why people universally make and listen to music, although it is highly costly and does not clearly promote survival or reproduction [44].

Chalmers [28] contrasted the “hard problem of consciousness” with corresponding “easy problems” such as how to explain human perceptual and cognitive abilities. These “easy problems” are the focus of regular empirical research in relevant branches of psychology, including the psychology of music (e.g., [45]), religion (e.g., [46]), and language (e.g., [47]). The present contribution aims to go further and “face up” (in Chalmers’ words) to more fundamental issues.

In the following text, various speculations about the origin of behavioral modernity will be evaluated. Depending on the question, different disciplinary perspectives will be adopted (e.g., developmental psychology, linguistics, musicology, religious studies, theology, philosophy, archeology, acoustics, anthropology). Different aspects of behavioral modernity (language, ritual, music, religion, reflective consciousness) will be considered separately. Characteristically musical and religious emotions will receive special attention. Diverse sources of evidence such as infant abilities, strong music experiences, and the content of religious texts will be considered. Finally, relevant epistemological biases of the author will be addressed.

## 2. Language

How and why did human communication become so complex? While language can probably be considered an evolutionary adaptation [48], neurocognitive accounts such Berwick and Chomsky [1] remain incomplete:
we do not really know how the Basic Property is actually implemented in neural circuitry. In fact … we don’t have a good understanding of the range of possible implementations for any kind of cognitive computation. Our grip on how linguistic knowledge of ‘grammars’ might actually be implemented in the brain is even sketchier (p. 157); “Basic Property” refers to the generation of a practically unlimited number of hierarchically structured expressions.

Hauser et al. [49] listed further unresolved problems:
(1) studies of nonhuman animals provide virtually no relevant parallels to human linguistic communication, and none to the underlying biological capacity; (2) the fossil and archaeological evidence does not inform our understanding of the computations and representations of our earliest ancestors, leaving details of origins and selective pressure unresolved; (3) our understanding of the genetics of language is so impoverished that there is little hope of connecting genes to linguistic processes any time soon; (4) all modeling attempts have made unfounded assumptions, and have provided no empirical tests, thus leaving any insights into language’s origins unverifiable. 

Harnad [50] asked “Where did our brains’ selective capacity to learn all and only UG-compliant languages come from?” (p. 524; UG = universal grammar) and emphasized the importance of avoiding circular reasoning, or begging the question:
We were looking for the evolutionary origin of the complex and abstract rules of UG. Christiansen and Chater [51] say … Don’t ask how the UG rules evolved in the brain. The rules are in language, which is another ‘organism,’ not in the brain. The brain simply helped shape the language, in that the variant languages that were not learnable by the brain simply did not ‘survive.’ This hypothesis begs the question of why and how the brain acquired an evolved capacity to learn all and only UG-compliant languages in the first place, despite the poverty of the stimulus—which was the hard problem we started out with in the first place!

Regarding the poverty of the stimulus—the idea that the speech to which infants and children are exposed is insufficiently complex to explain their rapid learning of linguistic details—data-driven statistical learning can achieve more than nativist approaches have assumed [52,53]. Statistical learning has also been observed in non-human primates [54]. Human infants imitate and participate in gestural-acoustic exchanges in a complex, embodied-interactive sensorimotor process [55], whose fine details involve far more “information” than a grammatically based cognitive-linguistic account.

Circular reasoning may be avoided by seeking an independent biological trigger that catalyzed genetic and/or cultural changes and eventually led to the emergence of complex, reflective language. Such a trigger should have nothing to do with language itself, nor should it involve any other aspect of behavioral modernity (technology, religion, art, music, consciousness) that may have interacted with language during the period when it was probably emerging.

### 2.1. The Obstetric Dilemma and Proto-Language

Human–chimpanzee speciation occurred at least six million years ago [56]. From around that time, the mother’s bipedal gait [57] and the fetus’s increasing brain size made the birth process (parturition) increasingly difficult [21]. Shortened gestation necessitated social support (midwifery) and rendered human infants more fragile [58]. 

While humans are not the only primates for whom birth is difficult and newborn clinging/crawling impossible [59,60], the situation is more extreme for humans than for other primates in both respects. Washburn [61] explained:
The human mother-child relationship is unique among the primates as is the use of tools. … In man adaptation to bipedal locomotion decreased the size of the bony birth-canal at the same time that the exigencies of tool use selected for larger brains. This obstetrical dilemma was solved by delivery of the fetus at a much earlier stage of development. But this was possible only because the mother, already bipedal and with hands free of locomotor necessities, could hold the helpless, immature infant. The small-brained man-ape probably developed in the uterus as much as the ape does; the human type of mother-child relation must have evolved by the time of the large-brained, fully bipedal humans of the Middle Pleistocene. Bipedalism, tool use and selection for large brains thus slowed human development and invoked far greater maternal responsibility. The slow-moving mother, carrying the baby, could not hunt, and the combination of the woman’s obligation to care for slow-developing babies and the man’s occupation of hunting imposed a fundamental pattern on the social organization of the human species. (pp. 73–74)

Higher infant death rates meant that infant and mother behaviors were more strongly selected for if they promoted infant survival. Infants and mothers became more sensitive to each other’s needs and states, communicating this information by increasingly complex combinations of vocalization, body movement, and touch—consistent with psychological theory of attachment [62]. Attachment is essential for infant survival across species [63]. A human infant’s attachment to its primary carer predicts the quality of adult relationships, which in in turn predicts future attachment to infants [64].

An explanation of this kind was introduced by Dissanayake [18,19,65]. Brown and Dissanayake [66] explained:
Such coordinated, dyadic behavior … addressed the “obstetric dilemma” of two million years ago when the anatomical trend toward a narrowed pelvis in fully bipedal Homo erectus conflicted at childbirth with a concomitant anatomical trend toward enlarged brains and skulls. Among other adaptations (e.g., separable pubic symphysis in females at parturition, compressible infant skull, extensive postnatal brain growth), the gestation period was significantly reduced, resulting in helpless infants dependent on their caretakers for years, rather than weeks or months as in other primates. A mother’s simplification, repetition, elaboration, and exaggeration of affinitive communicative behaviors (e.g., smiling, open eyes, eyebrow flash, head bob, head nod, soft undulant vocalization, touching, patting, kissing) served to reinforce affinitive neural networks in her own brain and, when performed on a shared temporal basis, also set up a means of neural coordination of behavior and of matching of affective change between the pair.

In many species, the risk of death is highest in the first weeks, months, and years of life. Hill and colleagues [67] estimated infant mortality in wild chimpanzees at “about 20% in the first year, dropping to a minimum of about 3.5% between ages 10–15” (p. 442). Today, avoidable global child mortality (mainly from disease and hunger) is about nine million per year [68]—much higher than the adult global death rate due to violence. Infant mortality is relatively high in hunter-gatherer societies [69]; 100,000 years ago, it may have been higher and possibly included infanticide [70,71]. 

The best survival strategy for an infant, human or otherwise, is to maintain maternal proximity and attention. When a human infant or young child is separated from its primary carer, and its initial protests are unsuccessful, its behavior becomes increasingly desperate and fearful; it loses appetite and cannot sleep [62]. Physiological measures of infant emotional reaction include piloerection (goose bumps, chills) and increased cortisol levels.

Infancy is characterized by high neuroplasticity [72]. In combination with high infant mortality, that makes the events and constraints of early developmental periods more relevant for natural selection than those of later periods [17]. The human infant brain also grows faster than in other primates [73,74]. These innate predispositions may have allowed the human fetus to become more sensitive to complex patterns of vocal prosody and other prenatally perceptible stimuli (cf. [75,76,77], which in turn enabled them to develop stronger maternal attachments and survive with higher probability. Then and now, more secure children are more linguistically competent [78].

In this way, early human infants may have formed higher-level associations that increasingly involved abstract concepts. Consider for example a child’s concept or schema of “dog”, which includes general characteristics such as four legs, different kinds of dogs, typical canine behaviors, and canine affordances from the child’s viewpoint. As in most words [79], the sound of the word “dog” is arbitrary and unrelated to conceptual content; the sound-meaning relationship must be learned. Sound patterns produced within the mother’s body and audible by the fetus also have learnable meanings. As the ability to learn large numbers of schemas/meanings (nouns, verbs) and join them in different combinations (dog barks, dog runs, girl runs) became crucial for infant survival, mothers and infants gradually developed this ability in parallel. A simple theory of this kind, combining evolution with associative learning, can account for the emergence of human linguistic ability. 

### 2.2. Motherese and Proto-Language

Motherese is a playful sing-song interaction between adults and infants. It involves temporal multimodal coordination of gestures, facial expressions, and sound patterns, to which infants are especially sensitive [80]. Motherese is not language, which emerges about two years later. But the ontogenetic emergence of language relies on perceptual, cognitive, motor, and social foundations that were laid in infancy and even before birth [81].

The uniqueness of modern human motherese suggest that proto-motherese played a role in the phylogenetic emergence of both music and language [82,83]. Sharing properties of both music and language, motherese corresponds well to Brown’s [84] *musilanguage*. Simple grammatical functions may have originated during motherese as mother and infant labeled objects (nouns), processes (verbs), interactions (prepositions) and so on (syntactic bootstrapping [85]). Bloom [24] pointed out that
All languages have a word that refers to hands, for instance, but this is probably because it is important for people everywhere to talk about hands, not because of a specific innate propensity toward hand-naming.(p. 148)

Complex patterns of vocal prosody have meanings such as “arousing/soothing, turn-opening/turn-closing, approving/disapproving, and didactic modeling” [86] (p. 415), of which approval versus prohibition [87] is crucial for infant survival. Complex sound-meaning relationships of this kind, and variations (playful or otherwise) in which the noun stayed the same and the verb changed or vice-versa, may have helped early human infants learn to combine syntactic functions (cognitive “merge”).

The process did not begin from scratch. Non-human primates communicate by means of complex multimodal combinations of gestural, vocal, and facial signals [88]. Primate infants are predisposed to learn such patterns and their meanings in trial-and-error processes of playful experimentation, usually in social-interactive settings. Early humans presumably did the same, but with a larger vocabulary and more complex grammar. Falk [17] explained:
Goodall [89] notes that vocal communication of chimpanzees is far more complex than previously appreciated, and has classified 34 discrete calls along with the emotions with which they are associated. She also observes that chimpanzee listeners learn much from the sequences of vocalizations that pass back and forth between individuals. (For example, the screaming of an adult followed by squeaks and then pant-grunts indicates to a distant chimpanzee that an aggressive interaction has occurred and that the victim has relaxed and approached the aggressor.) Chimpanzee calls are distinguished (with presumably more difficulty for human than chimpanzee listeners) from an acoustically graded continuum.(p. 492)

In a statistical-learning approach, the complexity and ambiguity of human grammar and vocabulary are consistent with, and explicable by, two factors. The first is the complexity of human neural networks—enabled by enlarged cortex, and possibly a result of social constraints in larger groups [90]. The second is the repetitiveness of prenatal sound patterns and infant-mother exchanges in motherese as they occur in real-world contexts; music is similarly repetitive [91,92]. The high speed at which the human fetus/infant learns arbitrary sound-meaning relations (cf. [93]) may have been necessary for survival in ancestral gatherer-hunter environments. Its byproduct was a lifelong process of human cultural learning.

## 3. Ritual

Found in all human societies, rituals involve the shared performance of traditional, rule-governed, symbolic, special (sacred), culture-specific action sequences. Ritual participants use role-play to affirm personal allegiances and group identity, which helps them respond cooperatively to changes in social environment (births, initiations, marriages, deaths, new leaders, conflicts) and physical environment (seasons, weather, droughts, floods, shortages, abundance). Rituals feature medical, psychological and social healing processes [94].

Ritual is not confined to humans. A general definition includes “those stylized displays reported by ethologists to occur among the birds, the beast and even the insects” [95] (p. 25). Eilam and colleagues [96] considered that “motor rituals are characterized by their close linkage to a few environmental locations and the repeated performance of relatively few acts” (p. 456). 

Diverse connections among religion, music, and ritual suggest that religion and music arose in the context of ritual [97]. Repetition of sound and movement patterns might have activated MS, given that prenatally perceptible stimuli are generally repetitive. The effect might have been amplified if the “environmental location” referred to by Eilam and colleagues was a resonant (booming) cave whose audible spectral energy distribution was similar to that of the uterus from the fetal perspective [98]. Music and religion may have emerged in this way. By contrast, language emerged in everyday life situations, consistent with its relative ubiquity and its status as an evolutionary adaptation.

Early human rituals presumably mixed (proto-) music (song and dance), spirituality (proto-religion and magic), and substance use (psychoactive drugs, entheogens; [99])—similar to modern rave culture [100]. As today, ritual participants shared strange, wonderful emotions such as awe (transcendence, a magical feeling of accessing different worlds) and fusion (a secure feeling of belonging to other people and things [101]). Unaware of the origin of these emotions, participants collectively invented (created) supernatural explanations in a multigenerational process of oral transmission.

### 3.1. Cave Acoustics

Ancient musical instruments have been found in caves (Germany, Slovenia [102]), and ancient cave paintings have been found at acoustically resonant locations [103,104]. A possible reason: the special optical and acoustical properties of caves may have activated prenatal MS. Optically, caves are relatively dark—similar to the prenatal situation. Acoustically, caves whose walls are a few meters apart reinforce sound energy near 100 Hz [105,106], corresponding to a wavelength of about 3 m. Ternström [107] observed that a resonant basement in which a choir held a rehearsal “had a cluster of prominent resonances around 200 Hz” (p. 19).

From the fetal perspective, the mother’s voice lacks higher harmonic partials, the amniotic fluid acting as an acoustic low-pass filter [98]. Pure tones and partials are usually weaker above 300 Hz than below; above 1000–2000 Hz, inaudible. Noise bands from fricative consonants may be completely masked. The most clearly audible tones lie in the range 100 to 300 Hz, corresponding to the fundamental frequency of a male voice or bass singer. 

Resonant caves have longer reverberation times than other settings where people talk, so the optimal rate of intelligible speech there is slower. That could have shifted the attention of listeners toward prosody (contour and emotion) and away from phonetic and lexical content (cf. [108]), especially when a phrase was repeated [109]. Repetition is a feature of religious indoctrination that helps participants remember ritual content [110]. 

Worldwide, religious rituals happen preferably in resonant spaces (churches, temples, synagogues, mosques), although speech intelligibility in such spaces is relatively low [111]. Many early Christian rituals occurred in underground catacombs that counterintuitively “could not be used for prayers, meetings and religious functions because the echoes would have created difficulties in concentration and mediation for the devotees due to the poor speech comprehension” (p. 583). Religious leaders nevertheless emphasize intelligibility—the importance of understanding sacred texts in ritual situations [112]. The contradiction can be understood if the primary aim of religious rituals to activate the participants’ MS, creating the impression of communication with supernatural agents:
The high levels of absorption at high frequencies do not favor verbal audibility, since these frequencies are fundamental for the understanding of the spoken word. This is therefore a space for liturgical music, composed to inspire religious contemplation. [113] (p. 311)

Ancient cave paintings suggest totemism: beliefs in spirits or sacred animals to which people are mystically related and with which they identify [114]. Why did early humans depict totems or spirits on cave walls at acoustically resonant locations? MS theory suggests that ritual participants attributed sound-evoked emotion to spirits within the rock—analogous to the mother as perceived by the fetus, outside and above the uterine wall. MS activation can explain why they anthropomorphized totems, ascribing minds or souls to them. Lacking acoustical knowledge, they could not otherwise explain this special ritual experience [104,115].

### 3.2. Changed States: Shamanism and Hypnosis

Shamanism arose independently in different cultural and geographical contexts [116]. Shamans engage in traditional practices to alter states of consciousness, enabling ritual participants to perceive and interact with the spirit world. The purpose is often to heal, and the ailment in question may be either medical, psychosocial, or both (the two often being inseparable). Shamanic states (spiritual possession, trance) are comparable with hypnotic states in the modern western tradition, despite large cultural differences [117]. 

Both hypnosis and shamanism are consistent with activation of MS. An infant’s survival strategy includes obeying the mother/carer [17] or guessing her/his intentions and following them [118,119]. Both the shaman and the hypnotist may be playing the role of the mother/carer as perceived by the prelinguistic infant, consistent with the ability of both to heal. 

### 3.3. Postures and Movements 

The behavioral organization of all mammals begins prenatally [120,121]. For example, human fetal hand movements are already goal-directed at 22 weeks [122]. During the second half of gestation, the hands of the human fetus often touch each other and the face or mouth; thumb sucking is also observed [123]; “the subtypes of hand to head movement are: hand to head, hand to mouth, hand near mouth, hand to face, hand near face, hand to eye and hand to ear” [124] (p. 496). Near term, fetal posture becomes more rounded and movements more limited for lack of space; the confinement is so extreme in humans that the mother risks unique pathologies [125]. 

MS theory predicts that MS is evoked in children and adults by body postures and movements that are similar to fetal postures and movements. Relevant behaviors include postures adopted in prayer and meditation in different cultures: bowing, kneeling (genuflection), prostration, sajdah, meditation postures, holding the hands together in front of the face, or touching the ground with the forehead. 

Alternatively, ritual postures can be explained by the power difference between a worshipper and her/his god, and the desire to depict and feel humility (e.g., [126]). In social dominance hierarchies, individuals make themselves smaller and more vulnerable by adopting closed or bent body positions (bowing, kneeling, prostration), for example when submitting to an alpha male or to a dominant female or male [127,128]. In meditation, a more upright position is adopted because appeasement of a personal god is not necessary or because bent postures cannot be maintained for long periods.

MS theory can also explain the universal human phenomenon of dance. The dancer perceives entrainment (coordination, synchrony, coupling, stable phase relationship) between musical sound and dance movements. Similarly, the fetus perceives entrainment between footstep sounds and fetal movements when the mother walks [129,130,131,132,133]. A link of this kind can explain the experience of groove, understood as a “pleasurable drive toward action” in which “measures of the quality of sensorimotor coupling predict the degree of experienced groove” [134]. 

### 3.4. Ritual Emotion

When a human community comes together in a ritual setting, emotions are shared, strengthening the group and feelings of belonging (*collective effervescence* [114]). Such meetings are perceived as *special* [65]—in part because they are different from everyday work:
One generally finds, even in animals, “rules” of play: special signals (such as wagging the tail or not using claws), postures, facial expressions, and sounds that mean “This is make-believe.” Often, special places are set aside for playing: a stadium, a gymnasium, a park, a recreation room, a ring or circle. There are special times, special clothes, a special mood for play—think of holidays, festivals, vacations, weekends. (p. 17)

Consider ritual rites of passage, in which the status of ritual participants changes. The temporary uncertainty (ambiguity, disorientation) of their status during the ritual (*liminality* [135]) coincides with changed states of consciousness, which invites ritual participants to assume a causal link between two. When several people share an intense liminal experience, they feel a heightened sense of bonding, social equality, and solidarity (*communitas* [135]). Even without such intense experiences, the “bandwagon effect” [136] means that if several people share the same idea, others are more likely to accept it. 

Skinner [137] demonstrated that behavior can be changed by systematic application of rewards and punishments. In an evolutionary approach, spirits or gods reward or punish certain behaviors, making group behavior more coherent and promoting the group’s survival in competition with other groups [138,139]. The emotions that are created and shared in ritual settings may act as rewards in Skinner’s approach, positively reinforcing behaviors that appear to have produced them. If music-like patterns of sound or religion-like behaviors induce positive emotions, these can act as rewards in a behavioral paradigm, and ritual participants will try to recreate them. 

Music evokes special emotions that differ from everyday emotions [140]. MS theory offers a possible explanation: ritual activities involve perceptual patterns that are similar to patterns perceived by the human fetus. The patterns therefore evoke MS and associated emotions. The emotions act as Skinnerian reinforcers and are shared by ritual participants, who collectively attempt to recreate the emotions. Despite this universal mechanism, human religions and musics are diverse due to the diversity of physical and social contexts in which human rituals take place.

There is considerable overlap between emotions evoked by MS, the infant schema, and other schemas or stimuli, suggesting these schemas might be related to each other. Emotions typical of drug-induced ecstasy, music, and social bonding are also similar to each other. Awe, for example, can be triggered by MS or interactions with a powerful alpha male. 

Two distinct behavioral settings are relevant for this argument: the mother-infant dyad, and rituals involving several/many adults/children. In mother-infant communication, the mother activates the infant’s MS and the infant activates the mother’s infant schema. In adult group rituals, the MS of each participant is activated by the collective behavior of the group. Adult ritual participants depend on each other just as infants depend on their mothers, but to a lesser degree. The link between the two cases may be an example of Piaget’s [141,142,143] assimilation and accommodation (cf. 31). Physiological systems for infant-mother attachment may be transformed or adapted to support the individual’s attachment to—and psychosocial identification with—family, clan, country, or other group. As rituals become more complex, behavior is increasingly co-determined by culture-specific ritual schemas that encapsulate explicit and implicit knowledge about ritual procedures.

## 4. Music

A theory of the origin of music should start from a definition of music. Music may be defined as “humanly organized sound” [144], but many humanly organized sound patterns are not music (e.g., machine noise). Music usually involves song and/or dance, but song and dance are themselves hard to define. Wikipedia claims that “Music is an art form and cultural activity whose medium is sound organized in time” (accessed 29 Sep 2018), but what are the definitions of “art form” and “cultural activity”? Philip Tagg’s teaching materials (tagg.org/teaching/musdef.pdf) propose that
Music is that form of interhuman communication in which humanly organised, non-verbal sound is perceived as vehiculating primarily affective (emotional) and/or gestural (corporeal) patterns of cognition. 

But this promising statement is more description than definition, and it raises an additional question: Why does music involve certain human senses (hearing, proprioception, balance, acceleration) and not others (touch, vision, smell, taste)? Music’s combination of hearing and proprioception seems arbitrary given that other combinations such as vision and hearing are possible. Remarkably, visual rhythms do not induce the same sense of movement as auditory rhythms [145]. A possible explanation involves the universal link between rhythm and dance; as the mother walks, the fetus moves rhythmically in time with the sound of footfalls. Similarly, melody may be based on maternal vocalization as perceived by the fetus. 

If that is true, music may be defined as sound/movement patterns that activate MS, producing characteristic emotions such as awe and fusion. But this kind of definition is explanatory rather than operational; it precludes an objective evaluation and does not allow a musicologist to decide subjectively whether or not given sound patterns are “music” without additional contextual information including social and psychological responses.

### 4.1. Music as Social Glue 

Music has diverse properties and functions [146,147], which explains in part why theories of music’s origin are so diverse [148]. A widely accepted idea is that music exists because it promotes social cohesion. According to Huron [149], “It may contribute to group solidarity, promote altruism, and so increase the effectiveness of collective actions such as defending against a predator or attacking a rival clan” (p. 47). Evidence for this *social glue hypothesis* includes “self-other merging as a consequence of inter-personal synchrony, and the release of endorphins during exertive rhythmic activities including musical interaction” [150]. The social function of music is not confined to humans; “our social–emotional propensity to occasionally gather in excited group displays is shared with our closest relative among the apes” (i.e., the chimpanzee) [151] (p. 6). If so, how are humans different?

Motherese in modern industrialized countries can hardly explain the origin of music if it was absent in ancient hunter-gatherer societies. Studies of modern hunter-gatherer cultures may be relevant here. Hewlett and Roulette [152] reported that “natural pedagogy was impacted by the Aka cultural context and interactions relied more on touch, physical proximity and pointing, and less on verbal exchange and motherese” (p. 10). But in an embodied cognition approach [153], touch (physical proximity) and gesture (e.g., pointing) are as important for motherese as vocalizations, and inseparable from them. Ancient proto-motherese presumably also involved emotional contagion and mutual empathy [154]: monitoring the emotional state of the other, joint attention, and imitation [155].

A theory of music’s origin that is based on the closest relationship of the lifespan (the mother-infant relationship from the infant perspective) is consistent with these observations. It explains in addition how music’s emotional and social functions might have arisen from its characteristic sound/movement patterns—universals of rhythm and melody that are linked to physical properties of the human body [132]. Most of the 70 putative musical universals listed by Brown and Jordania [156] are consistent with MS theory; Teie [157] attempted a systematic comparison. 

If music and religion arose together in ritual, how and why did they separate? Operant conditioning can explain how musical rituals might promote social cohesion without supernatural involvement, by evoking emotions typical of the mother-infant relationship from the infant perspective. As humans became aware of the stimulus-emotion relationship in such rituals, they started promoting non-religious music for its own sake. Dissanayake [158] coined the terms *ritualization* and *musification*:
music is conceptualized as a behavioral and motivational capacity: what is done to sounds and pulses when they are “musified” — made into music — and why. For this new view, I employ the ethological notion of ritualization, wherein ordinary communicative behaviors (e.g., sounds, movements) are altered through formalization, repetition, exaggeration, and elaboration, thereby attracting attention and arousing and shaping emotion. (p. 169)

### 4.2. Strong Experiences with Music

Social glue theory alone can hardly account for reports of extraordinarily powerful supernatural experiences in connection with music [43], monotheistic religion [159], or shamanism and spiritual possession [160]. Vivid illusions of supernatural encounters are reported in “peak experiences” [161], “transcendent ecstasy” [162], and “intense emotional responses to music” [163]. In a study of aesthetic experience that includes music, Panzarella [164] distinguished “renewal, motor-sensory, withdrawal, and fusion-emotional experiences.” Schäfer and colleagues [165] described “intense musical experiences” (IMEs) as follows:
(1) IMEs are characterized by altered states of consciousness, which leads to the experience of harmony and self-realization; (2) IMEs leave people with a strong motivation to attain the same harmony in their daily lives; (3) people develop manifold resources during an IME; (4) IMEs cause long-term changes to occur in people’s personal values, their perception of the meaning of life, social relationships, engagement, activities, and personal development.(p. 525)

Gabrielsson’s [43] respondents spontaneously reported powerful, surprising, unexpected spiritual or religious experiences when listening to or performing music. A singer reported a performance during which the roof above his head disappeared, stars and moonlight appeared, and he found himself inside his song. An audience member in a musical play felt she had become a spirit without a body (“it was a floating, weightless feeling”; p. 161). Reports of trance experiences by concert audience members and listeners to recorded music may involve floating, flying, or weightlessness:
The music was suddenly there round about me, as if it comprised a transparent but evidently impenetrable wall. I thought that it told me something, and I listened and answered, and when the music/story went on, I felt a joy that was so enormous that I experienced it as being almost cosmic. The condition that the story led me into was plastic—almost as if I was floating around or hovering inside the transparent wall. Nothing, nobody, could reach me. It was like a salvation, but without religious elements, and the warmth and joy and the calm that I experienced and heard long followed me. The experience influenced me for a long time after that in a deep and distinct manner.(pp. 163–164)
The feeling was mixed successively with a sort of elation of incorporeal floating, a total merging with the music, or quite simply with something bigger—God or the universe, perhaps—where the experience of me, myself, was completely annihilated. I think of it as that the experience at this stage is very like a religious salvation experience of being high on drugs.(p. 164)

Gabrielsson [43] reported many other such transcendent (otherworldly) experiences, for example:
“(…) a cosmic total experience beyond time and space. My body and the music became a whole where I knew I was dead, but it was a death that was a birth into something that was liberatingly light. A light that didn’t exist in this life. I even vanished from this life, so I can’t remember anything of my surroundings. Everything that happened wasn’t connected to this world”.(p. 168)

Music is often experienced as a virtual person or persona [166,167]. In reports of strong experiences, the persona may be identified with a religious god or spirit, and the encounter can be life-changing:
The song actually saved my life. Every time I hear the song I am filled with a feeling of joy. … I have been selected to live, I think that ‘someone’ saved me. Not God or any divine power. But this ‘someone’ whom I can’t place, they are just there. They are there like a big shadow, not only for me, deep inside, but in the whole of history, the whole history of life. I can only describe it by saying that it frightens me, but it makes me curious. It is magnificent and absolutely unique, but terrible and indestructible. [43] (p. 60)

Gabrielsson’s data challenge the rationalist cognitive paradigm of music psychology, suggesting that irrational or parapsychological aspects may be fundamental. If participants’ comments are evidence of prenatal MS being activated, which includes a feeling that another being is present (vaguely large, moving, above, and outside) or a feeling of weightlessness and floating in (amniotic) fluid, the respondents in such studies were unaware of any such connection. Instead, they explained their strong emotions and changed states in terms of adult relationships and situations: families, other groups and social identities, and musical/religious rituals. 

## 5. Religion

Like music, religion is difficult to define. Definitions often involve cultural behaviors and traditions that are based on beliefs in supernatural agents. Sosis [168] (p. 343) proposed that “although countless scholarly definitions of religion have been offered, […] “‘belief in supernatural agents’ might win a popular vote”. Singleton and colleagues [169] (p. 250) defined “spirituality” as a “conscious way of life based on a transcendent referent”. Rossano [170] defined religion as “beliefs or actions predicated on the existence of supernatural entities or forces with powers of agency that can intervene in or otherwise affect human affairs” (p. 346). The behaviorally costly features of religion depend ultimately on beliefs in supernatural agents: Sosis [168] (pp. 343–344) listed “ritual, myth, taboo, emotionally charged symbols, music, altered states of consciousness … and afterlife beliefs among others.” For Saroglou [101], believing (in gods, spirits, afterlife; awe, transcendence) was the first of four cross-cultural dimensions of religion, the others being bonding (ritual, love), behaving (morality, social emotions), and belonging (community, affiliation, identity).

The religions of the world offer diverse answers to two central questions: Where did the universe come from? What happens to the “soul” or “spirit” after death? The prehistoric emergence of language and reflective consciousness made such questions possible. We might suppose that both questions were posed by all ancient peoples. In pre-scientific culture, it seemed obvious that the soul left the dead body and continued to exist in some form. To explain the origin of the universe, it is not surprising that ancient peoples came up with creative agents (creators) and creation myths.

What is surprising about the world’s religious cultures is the widespread counterintuitive belief in a human or animal creator, although the universe is obviously far too big for a human or animal to create [171]. According to Epley and colleagues [172], “people are more likely to anthropomorphize when anthropocentric knowledge is accessible and applicable, when motivated to be effective social agents, and when lacking a sense of social connection to other humans” (p. 864). But that can hardly explain why diverse early human societies—already capable of making and using tools, building shelters, burying the dead, and creating art—quasi-universally and quasi-independently supposed that an animal or human-like entity could create the stunning vastness of the earth’s landscape and the night sky, and that they could at some level communicate with the creator.

Most gods in human history are humanoid; even the exceptions retain humanoid features. The ancient Chinese concept of Tao (Dao) represents life or liveness (on earth), but at the same time fills the universe. It is the fundamental force or energy behind everything, everywhere and at all times, but intimately connected with human life and morality. The ancient Egyptian creator-sun-gods Aten and Ra combined natural features (the sun, or rays of sunshine) with humanoid expressions (masculinity, or a male-female mixture). The god of Sikhism (Vāhigurū) is not humanoid, nor can it be described; but it can be seen by the human inner eye and felt from the human heart during meditation. The belief in human- or animal-like supernatural agents with whom humans can communicate is simple, unique, and universal—inviting a parsimonious explanation (cf. [173]). 

### 5.1. The Experience of Divine Presence

Existing theories of the origin of religion have difficulty accounting for the vividness and detail of accounts of divine communication, and the frequency with which such encounters are reported. The 2004 US General Social Survey found that 37% of people experience “God’s presence” many times a day or most days; 22% never or almost never experience this [174]. Hay [175] reported that over 1/3 of adults in Britain and the United States and 2/3 of a random sample of postgraduate education students reported having had mystical, transcendental or paranormal religious experiences involving god or a supernatural power. Usually, the power was humanoid and the respondent was alone when it happened. The experience lasted for a few minutes and provided relief from preceding distress. Reported feelings included peace, ecstasy, and joy. Many thought the experience made them a happier or a better person. 

Are these experiences examples of *perception* in the psychological sense of “organization, identification, and interpretation of sensory information in order to represent and understand the presented information, or the environment” [176]? From a Christian perspective, Alston [159] argued that “experiential awareness of God, or as I shall be saying, the *perception* of God, makes an important contribution to the grounds of Christian belief” (p. 1, italics in original). If so, the subjective reports of religious respondents should be taken seriously by psychologists. Similarly, in the philosophical theory of color perception, one cannot “understand” red without experiencing it directly [177]. 

A possible psychological explanation is that stimulus patterns in ritual situations or everyday life resemble those “experienced” before birth in connection with the mother. The stimuli then trigger MS, creating an illusion of divine communication. Christian expressions such as “being filled with the holy spirit” and the “glory (splendor, magnificence, wonder) of God” are consistent with this idea.

Divine presence is regularly experienced by both religious professionals (ministers, preachers, etc.) and lay people. Expert reports may be more detailed and consistent [178] but also more biased by scriptures (religious texts) and proselytism. Alston [159] (pp. 12–13) presented examples. Consider the following three, beginning with a (non-expert) respondent from James [179] (pp. 67–68):
… all at once I … felt the presence of God—I tell of the thing just as I was conscious of it—as if his goodness and his power were penetrating me altogether.… I thanked God … I begged him ardently …. I felt his reply … Then, slowly, the ecstasy left my heart; that is, I felt that God had withdrawn the communion which he had granted … But the more I seek words to express this intimate intercourse, the more I feel the impossibility of describing the thing by any of our usual images.

Another report in James [179] (p. 250) alluded to liquid or floating:
… the holy spirit descended upon me in a manner that seemed to go through me, body and soul. I could feel the impression, like a wave of electricity, going through and through me. Indeed, it seemed to come in waves and waves of liquid love; for I could not express it any other way.

The following statement by the 13th-century Italian Franciscan tertiary Angela of Foligno [180] (see [179], p. 68) is similarly consistent with the idea of MS activation:
At times God comes into my soul without being called; and He instills into her fire, love, and sometimes sweetness; and the soul believes this comes from God, and delights therein. But she does not yet know, or see, that He dwells within her; she perceives His grace, in which she delights … For the eyes of the soul behold a plentitude of which I cannot speak; a plentitude which is not bodily but spiritual, of which I can say nothing. 

In a qualitative study of religious subjectivity, Kaplan [181] “examined the experience of seeking, receiving, and following guidance from a perceived source of divine wisdom”. Participants were experts (“advanced spiritual teachers”; p. iv). Forms of divine communication were classified into inner voice, channeling, intuition, inspiration, and synchronicity. The following comments are consistent with the thesis that concepts of god and divine communication originated in the mother-infant relationship:
My belief is that it is the ground of being, that it is a spirit that holds creation … something that’s very, very immediate … it’s more here than I am here right now. … To me this divine is someone, some being, who knows me and loves me … It is someone who can address me and guide me, whom I can trust and don’t need to be standoffish with or fearful of, because there’s an empathic connection. I feel that whoever this divine is knows me better than I know myself.

Alston proposed that such reports imply a direct experiential awareness of God and are therefore evidence for God’s existence. While they are surely distorted by religious knowledge and intentions, their prevalence, consistency and authenticity make it difficult to dismiss them as entirely fabricated or imagined. 

### 5.2. Existing Theories of Religious Origins

Research on both the origin of religion and the origin of music has a long, complex, speculative history. In both cases, researchers have tried to explain the psychological, social, and medical benefits of a universal, powerful, emotional, culturally diverse phenomenon. Both projects were hampered by the difficulty of separating nature from nurture. Both emphasize the importance of social cohesion in ritual [18,138,182,183,184,185]. Both may be avoiding a “hard problem”, namely the fundamentally transcendental nature of both religious and musical experience.

Is religion an evolutionary adaptation or byproduct [138,168,184]? Did religion arise as a pro-social adaptation that solves the free-rider problem? An extreme display or personal sacrifice in the context of a ritual is costly and hard to fake [185]. Ritual must therefore be honest. A ritual that requires such behavior promotes group cohesion; it is pro-social. However, religion does not necessarily promote group survival; it may even suppress innovation [186].

Theory of mind is consistent with divine belief but does not necessarily imply it. Theory of mind distinguishes humans from other animals, with some overlap [187]. Emerging in infancy [188], it enables humans to quickly infer the intentions of dangerous humans or animals, enhancing their ability to survive [189]. “Whether religion was inevitable or not is debatable, but it seems certain that only when human cognition allowed for the interpretation of one’s own mental state, could belief in supernatural agents become possible” [190] (p. 20). 

Johnson and Bering [139] explained the universality of supernatural beliefs as a combination of “(1) the selection of human psychological traits for monitoring and controlling the flow of social information within groups; and (2) attributions of life events to supernatural agency”. An explanation of this kind is convincing only if divine rewards and punishments are more effective than their human equivalents. But if people believe a priori in divine agency, the theory becomes circular.

Cognitive byproduct theories are promising. To reduce the chance of attack by dangerous humans or non-human animals, people overattribute agency. A *hyperactive agent detection device* attributes agency to natural events such as storms, which can in turn explain widespread beliefs in gods and spirits [191]. Another cognitive byproduct theory involves *minimally counterintuitive concepts* [192]. Religious stories are often intuitive in most respects and counterintuitive in only some. That makes the stories easier to remember and transmit in oral tradition. *Dual inheritance theory* combines cognitive byproducts (explaining supernatural agency) with prosocial adaptations (explaining cultural traditions) [138]. 

Theories of this kind explain important aspects of religious belief but cannot directly account for universally encountered counterintuitive beliefs in the creation of a large or infinite world by a relatively small or finite being, or reports of personal encounters with such creators or other supernatural agents. While monotheism is not relevant to theoretical explanations of the origins of musical or religious ritual, given the dominance of monotheism among today’s religions, it is interesting that MS theory is consistent with universal features of Allah in Islam or God in Judaism or Christianity (omnipresence, omniscience, omnipotence, omnibenevolence). Existing cognitive theories may be *necessary but insufficient* to explain universal attributes of monotheistic gods. 

### 5.3. Religious Emotion

Highly educated people may adhere persistently to a faith that includes supernatural agency—in the absence of direct sensory evidence and despite clear contradictions [193]. Perhaps the emotional drivers in favor of belief in the face of mounting evidence to the contrary are stronger than the opposing cognitive drivers. Cognitive byproduct theories can hardly account for the emotional force that drives religious belief:
Here the poverty of stimulus could not be more extreme, nor could religious responses be more robust. Consider adolescent Khoisa males in Southern Africa who endure excruciating ritual circumcision only to live in exile in a desert environment without any food or water until they heal. The initiates risk infection, dehydration, exposure, and willingly submit to certain agony. The Khoisa claim the gods demand this ordeal of them. But how can chopping bits of genitals before the heavens improve survival? [194] (p. 656)

Evolutionary arguments that attempt to explain the origin of strong religious emotion can fall into the trap of circularity (cf. [31]). Alcorta and Sosis [97] argued for “conditioned association of emotion and abstract symbols … the brain plasticity of human adolescence constitutes an ‘experience expectant’ developmental period for ritual conditioning of sacred symbols” (p. 323). But if “the meaning of abstract religious symbols must be created, both cognitively and emotionally” (p. 332), these symbols are previously unemotional. In that case, there is no external source of emotion with which to explain religious emotion. What if the emotion comes first and the abstract symbols emerge later?

### 5.4. Religion and Infant-Carer Attachment

The strong attachment of a human infant to its mother or carer [62]—assumed here to be a consequence of the obstetric dilemma—can explain aspects of religious experience. Granqvist and Kirkpatrick [195,196] and Granqvist et al. [197,198] presented systematic evidence for such a connection and considered its implications. According to Kirkpatrick [199],
The perceived availability and responsiveness of a supernatural attachment figure is a fundamental dynamic underlying Christianity and many other theistic religions. Whether that attachment figure is God, Jesus Christ, the Virgin Mary, or one of various saints, guardian angels, or other supernatural beings, the analogy is striking. The religious person proceeds with faith that God (or another figure) will be available to protect and comfort him or her when danger threatens; at other times, the mere knowledge of God’s presence and accessibility allows him or her to approach the problems and difficulties of daily life with confidence. (p. 6)
Bowlby [62] identified three classes of stimuli hypothesized to activate the attachment system: (a) frightening or alarming environmental events; (b) illness, injury, or fatigue; and (c) separation or threat of separation from attachment figures. If God functions psychologically as an attachment figure, then we should find that people turn to God, and evince attachment-like behaviors toward God, under these conditions. Indeed, in Western Christian traditions at least, these are precisely the three categories of “trouble and crisis” when people are most likely to seek God’s support and comfort. (p. 7)
To the extent that God functions psychologically as an attachment figure, we might expect the structure of individual differences in God images to resemble that of parental images. … In virtually every factor-analytic study published, irrespective of the particular kinds of items used, the first (and large) factor to emerge invariably reflects the idea of God as loving, caring, and benevolent. (p. 10)

### 5.5. A Scenario for the Origin of Proto-Religion

These considerations allow for a new, psychologically oriented theory of the origin of religion. Resilient beliefs in approachable humanoid supernatural agents may have two psychologically interdependent foundations: emotional and cognitive.

The emotional foundation may be the mysterious, emotional, virtual image of the mother as perceived by the fetus and infant. This image is evoked when MS is activated in religious rituals. Participants report the presence of a mysterious entity that is larger than themselves. The person experiencing spiritual/divine presence/possession is not (episodically) “remembering” a prenatal event in the everyday sense; instead, her/his real-time experience is being colored by subconscious prenatal associations. The associations are salient and the “memory” is strong due to its intrinsic emotionality (cf. [200]).

The cognitive foundation involves questions about creation and death that are inherently difficult or impossible to answer [24]. Each question is linked to an emotion: creation with wonder [201] and death with anxiety [202]. These questions and the strange feelings that are experienced when MS is activated are separate, but since both are inexplicable, early humans linked them.

### 5.6. Male Dominance

If gods and spirits are cultural transformations of the mother as perceived by the fetus and infant, why are gods are usually male? First, social structures are usually patriarchal; a male god is one of many consequences of male dominance [203]. Second, the fetus and infant have no concept of gender. 

Infants do prefer high-pitched speech and singing [204], faces that adults consider attractive [205], and adults that smile and maintain eye contact [206]. They are sensitive to the size of strangers (children versus large/small adults), upon which fear responses depend [207]. Gender differences in infants (under one year of age) have been demonstrated in visual interest for toys [208] and people versus machines [209]. 

But there is no evidence that infants have a concept of gender or understand gender distinctions, even though parents or other adults may treat baby girls and boys differently [210]. Verbal gender distinctions in infants emerge at around 21 months [211], after which girls start to exhibit greater empathy than boys [212]. 

## 6. Reflective Consciousness

The word “reflective” implies the experience of introspection, assumed to be confined to humans. The concept of (self-)reflective consciousness was operationalized in psychological development by Zelazo [29]. It involves theory of mind [213] and mental time-travel (imagining the past and future [214]). Both aspects are practically unique to humans [215,216]. In ontogeny, both emerge during mother-infant interactions [217], making those interactions promising candidates for a causal origin for consciousness. 

Infants and children are remarkably good at imitating the actions of others while at the same time guessing their intentions [218]. In a theory of the origin of behavioral modernity based on mother-infant interactions, reflective consciousness emerged during those interactions. Insofar as their success is crucial for infant survival, reflective consciousness may be considered an adaptation. Later, it continued to develop in the context of language, religion, music, and other arts. Mother-infant exchanges constructed the self (mind) by playful labeling and mirroring of intentions, evaluations and subjective experiences [219]. Toys to which infants and children feel attached were anthropomorphized [220], which motivated talking about them, contributing to linguistic skills. 

Humans are more likely to reproduce successfully if they can predict infant accidents that could be fatal, and prevent them. A carer must therefore constantly monitor and control the infant, while at the same carrying out other work that is necessary for individual and group survival, such as gathering food (“multitasking”). Control from a distance can be achieved by vocal prosody. Falk [17] argued that
mothers increasingly used prosodic and gestural markings to encourage juveniles to behave and to follow … mothers that attended vigilantly to infants were strongly selected for, and that such mothers had genetically based potentials for consciously modifying vocalizations and gestures to control infants.(p. 491)

An infant’s natural playfulness and curiosity can suddenly and with little warning create a life-threating situation. Modern parents have trouble predicting and preventing dangerous infant falls [221]. To predict the future based on past experience, they need considerable experience, knowledge, and imagination. The complexity of this process and the potentially fatal consequences of failure are consistent with the hypothesis that reflective consciousness emerges both ontogenetically and phylogenetically from the carer-infant relationship.

## 7. Prenatal and Perinatal Psychology

A theory of the origin of behavioral modernity that is based on prenatal psychology and the mother-infant bond can be evaluated in part by considering empirical studies of aspects of fetal/infant behavior, perception, cognition, and emotion. The third-trimester human fetus perceives in all senses and processes information about its environment like any other organism. In an ecological approach, perception is always active: the fetus attends to, interacts with, and extracts information from its environment [121,222,223].

### 7.1. Transnatal Memory

Memory is information storage that influences future behavior and experience. Transnatal memory is demonstrated by postnatal behavioral effects of prenatal perceptual inputs [224,225,226,227]. Transnatal memory is procedural (implicit), not episodic; the infant does not “remember” birth or any preceding or following event until it acquires simple language. But given the importance of memory for survival as the newborn rapidly and interactively discovers its world, and the need for all cognitive systems to function at birth to reduce the probability of infant death, transnatal implicit memory is predicted in an evolutionary approach to be robust. 

When considering fetal perception and transnatal memory, there is no need to consider reflective consciousness, which develops in the first few postnatal years [29]. In both human and non-human animals, most learning occurs in the absence of consciousness [228].

### 7.2. The Perceptible Prenatal Environment

The prenatal soundscape includes the mother’s heartbeat and respiration (constantly) and her voice and footsteps (for long periods daily). All such stimulus patterns depend on the mother’s physical and emotional state and may correlate (with time delays) with the biochemical content of placental blood and amniotic fluid [229]. The fetal brain is likely processing all such information, given that “auditory cortex is more adaptive to womb-like maternal sounds than to environmental noise” [230] (p. 1). The fetus also hears and processes musical signals [231], but that phenomenon is not relevant for a non-circular theory of music’s origins and will not be considered here.

The fetus perceives sound and movement for 20 weeks before birth [232]. The third-semester fetus is asleep roughly 90% of the time but with large individual differences [233]. The total waking time in this period is thus well over 200 hours. Information acquired during wakefulness is processed during active sleep [234]. 

Sheep are anatomically and physiologically similar to humans. In that case, Abrams and Gerhardt [235] observed that
the acoustic environment of the fetus is composed of continuous cardiovascular, respiratory, and intestinal sounds that are punctuated by isolated, shorter bursts during maternal body movements and vocalizations. The distribution of sounds is confined to frequencies below 300 Hz. (p. S31)

From the human fetal perspective, most audible sound and movement patterns represent the mother upon whom its survival depends. In an evolutionary approach, high rates of infant mortality in hunter-gatherer societies suggest the fetus will assign a considerable proportion of available resources to the perception and cognition of such stimuli, insofar as they promote later mother-infant bonding. 

### 7.3. The Mother and Infant Schemas

Living organisms interact with their social and physical environments by recognizing patterns that are organized into schemas. Activated by learned or innate sensory patterns, cognitions, emotions, and behavioral interactions, schemas facilitate interaction by assigning objects, behaviors, or situations to categories that remain stable despite environmental changes [236]. Cross-modal interaction is the rule rather than the exception [237]. Every object or situation with which an organism interacts has schematically organized ecological affordances. Schemas enable learning and psychological development [242—244]. New experiences are either incorporated into existing schemas (assimilation) or the schemas adapt to new experiences (accommodation). Perceptual and behavioral patterns are interconnected within schemas, of which MS may be the first to develop.

The *infant schema*—also called *Kleinkindschema* [238], baby schema, cute schema, or infant appeal [239]—is activated in adults by an infant’s multimodal “cute” appearance and behavior, including facial expressions and vocalizations. It motivates maternal protecting and nurturing behaviors, which promotes infant survival. Cuteness is not confined to appearance (e.g., relatively large eyes and round head) but also includes childlike behaviors such as when a child “is surprised innocently … looks into my eyes with a mischievous smile … hides himself/herself in shame … toddles” [240] (p. 1090). Infants also modulate their behavior depending on the situation (e.g., crying as deception rather than extortion [241]). In these ways, human infants actively promote infant-carer bonding in the longer term—beyond the more immediate effects of birth hormones such as oxytocin. 

The strong relationship between mother/carer and infant (infant-caregiver attachment) and the active role of both [119] suggest that the infant has an analogous *mother schema* (MS) [131]. Since Lorenz’ early work on imprinting in goslings, there has been surprisingly little interest in this multimodal cognitive representation of the mother from the fetal or infant perspective. Linked to appropriate behavioral responses, and activated by pre- or postnatally learned or innate perceptual patterns including facial expressions [242], MS incorporates feelings about the mother—just as the infant schema incorporates feelings about the infant. For older children, MS includes typical characteristics of “good” and “bad” mothers and from the child’s perspective [243]. Part of the “symbiotic bio-affective (mother-fetus) dyad”, MS may be “absolutely indispensable to harmonious growth” [244] (p. 208).

The MS of the infant and the infant schema of the mother reinforce each other. The MS of the infant is activated by dynamic, multimodal perceptual patterns including sounds (“baby talk”), sights (facial expressions), tactile sensations, smells, tastes, and emotions [155]. The infant schema of the mother is motivated by infant appearance and behavior. Both mother and infant schemas are assumed adaptive, enhancing the probability that the infant will survive to reproductive age by promoting mother-infant attachment. The fetus acquires MS to facilitate postnatal mother-infant interaction, and mother and infant schemas interact in motherese. Taken together, the two schemas may be considered the foundation of carer-infant attachment and love. 

Mother and infant schemas are examples of *person schemas* [245] or *personal constructs* [246] that operate consciously or unconsciously to organize thought, mood, and interpersonal behavior. Other-schemas tend to be biased toward self-schemas: people tend to be more interested in what others have in common with them than with differences [247]. Also relevant is the *relational schema* [248] and spontaneous attraction toward potential sexual partners.

Mother and infant schemas promote well-being. An infant is happier in the company of its primary carer. An adult couple’s happiness usually increases after the birth of their first child—more so for the mother than the father [249]. In motherese, infants prefer happier sounds [250]. Similarly, group (choral) singing increases well-being [251], as does weekly public religious activity in the USA [252].

### 7.4. Prenatal and Perinatal MS

The human fetus is repeatedly exposed to maternal sound patterns (voice, heartbeat, footsteps, digestion [77,253,254]) and movement patterns [223]. The early development of olfactory/gustatory (from 8 weeks gestational age), vestibular/cutaneous (11 weeks), and vestibular/auditory (20 weeks) senses [244,255,256,257,258] enables the fetus to learn and subsequently recognize perceptual patterns of this kind. 

The fetus can perceive maternal vocal prosody, which is related to maternal physical and emotional state and therefore existentially relevant [259]. Of all the physical signals that are available to the fetus for receiving information about the mother and communicating with her, fundamental frequency contour (speech prosody) may transmit the most useful information in the shortest time (cf. [260]). Prenatal learning can therefore explain the extraordinary sensitivity of human newborns to maternal emotional prosody, as well as the dependence of infant crying patterns on maternal language [261]. Vocal learning of this kind, while not confined to humans [262], evidently plays an important role in human language acquisition.

The mother as perceived by the fetus may be considered the first “object” in the sense of “object relations” [263,264]—hence the double entendre “mother of all schemas” [243]. Like any other perceptual schema, prenatal MS involves gestalt (perceptual grouping, figure-ground organization)—an early example of schema-based segregation in (auditory) scene analysis (cf. [265]). Foreground stimuli or “signals” (maternal voice, heartbeat, footsteps, digestion) are segregated from background stimuli or “noise”. Prenatal perception of the mother may be the earliest “perceptual parsing problem” [265] in ontogeny and the most critical for (human) survival; “temporal coherence” may allow different stimuli to be linked to each other to construct the MS. For example, as the mother’s state of arousal changes, patterns perceptible in sound (voice, heartbeat, footsteps) and movement change in partly corresponding, predictable, or invariant ways. 

MS is subject to rapid accommodation at birth, as prenatal MS transforms into postnatal MS (cf. [266]). “Delivery constitutes an important structural modification of the dyad, so that it is important to be able to reestablish it in the most precocious and best possible manner so as not to interrupt the mother-child bond” [244] (p. 208). In this regard, human infants are more flexible than non-humans, shifting attachments or developing multiple attachments [267]. 

Perinatal MS accommodation is related to filial imprinting in non-human animals: a rapid, intensive learning process during a critical period that is almost independent of physical/social context and reduces fear [268]. Imprinting stimuli are primary reinforcers that innately elicit filial behavior [269] and may be prenatally predisposed [270]. For example, incubator-hatched geese attach themselves to diverse moving stimuli within half a day of hatching [238].

### 7.5. Biochemical Foundations

Endogenous opioids (including endorphins and morphine) mediate, reinforce, and regulate the mother-infant bond and may underlie MS’s positive emotions [271,272,273,274]. The mother’s presence, warmth, nourishment, and multimodal communication (auditory, tactile, gustatory) stimulate the release of opioids in the infant, which have a calming effect and reinforce attachment. 

The neuropeptide oxytocin is primarily associated with parturition [275] and maternal behavior including lactation, suggesting that its main function is to activate the mother’s care for the infant after birth. Oxytocin also plays a role in later intimate relationships (primary social affiliations), creating an emotional sense of safety [276]—similar to opioids [272]. Oxytocin and vasopressin facilitate attachment in monogamous non-human species [277]. 

Opioids and dopamine are also functional in intensely pleasurable music experiences [278]. Dopamine is linked to more positive-euphoriant emotions, whereas opioids are more tranquilizing. The biochemical substrates of infant attachment might therefore explain aspects of adult ritual behavior:
the social connotations and activation of the endogenous opioid system become cross-conditioned during early ontogenesis, so that later in life whenever the opioid system is activated by stress and pain, social connotations could arise together with the paradoxically occurring euphoric states and, vice versa, opioid-mediated euphoric and trance-like states are enhanced by social affiliation. The need for and the possibility of identification are interwoven at a psychobiological level: regression promotes endogenous opioid mediation while endogenous opioids mediate affiliation, and help depersonalization by loss of ego boundaries.[94] (p. 79)

Rates of infant mortality in ancient societies were high, consistent with the salience and complexity of mother and infant schemas and their strong biochemical foundation. On that basis, we might expect MS to remain functional long after infancy, even if it no longer contributes directly to survival or reproduction. MS-activation may strengthen and thereby benefit other relationships, and hence individual and group survival and reproduction. If so, MS in children and adults is not vestigial, like the human appendix, but instead contributes positively to all human relationships throughout the lifespan. Mother and infant schemas and associated altruistic behaviors that “maximize inclusive fitness through the care of helpless offspring” [279] (p. 1305) may underlie all loving human relationships, given that both altruistic responding and offspring care require
(a) participation by nonmothers, (b) motor competence and expertise, (c) an adaptive opponency between avoidance and approach, and a facilitating role of (d) neonatal vulnerability, (e) salient distress, and (f) rewarding close contact. Physiologically, they also share neurohormonal support from (g) oxytocin, (h) the domain-general mesolimbocortical system, (i) the cingulate cortex, and (j) the orbitofrontal cortex. [279] (p. 1305)

Oxytocin and vasopressin are also involved in the monogamous behaviors (affiliation, pair bonding) of non-human animals such as prairie voles [277]. Similar principles apply to non-human primates, other mammals, and other eusocial species [280]:
Even species that are highly divergent from mammals, such as squid, crocodiles, clownfish, and rattlesnakes demonstrate functionally similar behaviors to sequester and protect young from predators during their most vulnerable developmental stage, shortly after birth.[279] (p. 1314, citing [281])

## 8. Psychology of Religious and Musical Emotion

Both religious and musical behaviors are emotionally motivated; conversely, both religion and music can function as emotional regulators [282,283]. Can an analysis of religion- and music-specific emotions shed light on the nature and origin of these central aspects of human behavioral modernity? 

In an everyday, non-psychological definition, emotion is “an affective state of consciousness in which joy, sorrow, fear, hate, or the like, is experienced, as distinguished from cognitive and volitional states of consciousness” (dictionary.com, 19 Sep 2018). Here, emotion is primarily a subjective experience (although it is also a physiological state). It may be brief or long-lasting, weak or strong, and does not necessarily focus on a specific object.

Basic emotions have clear ecological-evolutionary functions and are shared by non-human animals. Panksepp and Watt [284] listed seeking, fear, rage, lust, care, panic/grief, and play. Basic emotions expressed by music include anger, fear, happiness, sadness, and tenderness [285]. 

Self-conscious emotions such as shame, guilt, embarrassment, pride, and envy are primarily human, although possibly expressed by non-humans [286]. Developing during the second postnatal year (perhaps earlier [287]), they regulate social behavior and promote group cooperation—interesting for adaptive theories of the origin of religion/music. They are difficult to recognize purely acoustically [288] or visually [289]. 

The emotions experienced in religious-musical rituals can be divided into two kinds: everyday (basic and self-conscious) emotions, and special emotions that are characteristic of rituals. To understand the nature and origin of such rituals, we need to understand their special or characteristic emotions [290,291,292]. They include the categories *awe* and *geborgenheit*.

### 8.1. Awe

Transcendence is integral to religion. Institutional structures and personal orientations include transcendental elements in all cultures [293]. Feelings of transcendence can involve profundity, sublimity, magic, enchantment, mystery, awe, amazement, wonder, fascination, chills, trance, ecstasy, spirit possession, spirituality, reverence for a supernatural being or divinity, ego loss, oneness with the universe, experiences of other worlds, nostalgia/sentimentality for a remote idealized situation, perfection or triumph that cannot be achieved in everyday life, majesty, grandeur, magnificence, or exceptional beauty.

Awe—a mixture of wonder and fear—is felt in the (subjective) presence of a spirit/god. It refers to “perceived vastness … threat, beauty, exceptional ability, virtue, and the supernatural” [290]. Otto [294] called an experience that simultaneously evokes fear and fascination “numinous”. Music can evoke awe in non-religious situations, especially for those who score high on personality trait “openness to experience” [295]. The more general emotion of “feeling moved” or “being touched” is linked to awe [291]; as supernatural beliefs are central to religion, awe may be central to music: “Being moved and aesthetic awe, often accompanied by thrills, may be the most genuine and profound music-related emotional states” (p. 115). Maslow [161] considered “peak experiences”, characterized by blissful, ecstatic emotion, a loss of orientation in time and space, a feeling of warmth and cosmic unity, a feeling of effortlessly and expressively achieving one’s full potential without fear or doubt, openness to creative ideas, and living in the present moment. Csikszentmihalyi [296] studied how emotions of this kind can promote flow and creativity.

Awe is induced differently in different cultural contexts, but the process is often consistent with the thesis of MS activation. In the Xangô possession cult in Brazil, participants experience strange, uncanny feelings and chills (goose-bumps and shivers) [297]. Altered states of consciousness or trance are achieved by “austere conditions such as strict fasting and thirsting, forced strenuous exercise, seclusion, hyperstress with feeling of terror, inducing of pain, temperature and kinetic stimulations” [94] (p. 72). Techniques to promote spirit possession include “songs, invocations, objects and substances” and more generally “archetypality, rigidity, regularity, redundancy and spatial and temporal delimitation” [297] (p. 177). Spirit possession rituals may involve either executive possession (the transformation or replacement of identity) or pathogenic possession (spirits that cause illness and misfortune) [298]. In independent and geographically distant cultures, music is combined with “smells, flavours, touch, images, sounds, body techniques, memories, evocations” [297] (p. 186). Complex belief systems and narratives explain these experiences.

Psychoactive substances can alone provoke transcendental experiences. MacLean and colleagues [299] investigated effects of the hallucinogen psilocybin mushrooms. Participants who remembered taking an active dose reported having had a “profound, personally meaningful experience” (p. 726). Their descriptions suggested “classic mystical experience” (p. 721). Given an appropriate context or situation, music can also evoke changed states alone, in the absence of drugs [43]. Whether those states are caused by rhythm, chant, or some other aspect of musical structure, is unclear [99]. Laski [162] observed that ecstasy can be triggered by art (music, poetry, literature, drama, ballet, film) or specific non-artistic stimuli (sudden or flashing light, water or waves). 

The term *oceanic feeling,* proposed by Romain Rolland in the 1920s [300] was an attempt to describe a special emotion that is experienced in religious rituals across cultures. The subject experiences the universe as one: a unity and infinity of time and space, with herself or himself is an inseparable part. Rolland thought oceanic feeling could explain the universal motivation to pursue religion; Freud [301] argued that the infant does not differentiate itself from other people or the outside world, oceanic feeling being a regression into this infantile state. Similarly, the Australian aboriginal concept of *dreamtime* or *dreaming* is a feeling of belonging to a universe in which ancestors are always present and can be met and communicated with, sometimes in song and story [302]. The magical creatures of the dreaming are at once timeless, spiritual, and everyday [303]. The Western concept of waking dream (hypnagogia) or lucid dream may be compared with the tantric concept of *dream yoga* and related phenomena in native North American culture that “provide a cosmic doorway into another dimension of reality” [304] (p. 183). 

### 8.2. Geborgenheit

Geborgenheit is another emotion that is typical of both music and religion. This German word, often considered untranslatable, alludes to feelings of physical safety and protection, emotional security, union (fusion) with another person or other people, and being loved or valued unconditionally. It implies closeness, warmth, snugness, cosiness, intimacy, stability, shelter, and peace. Infant primates tend to gravitate toward a single carer (monotropy [62]); geborgenheit is what we imagine they feel in contact with the carer’s warm body [305].

Geborgenheit is adaptive if it motivates infants to strive for and maintain their personal safety through attachment (cf. [62]). The mother regulates the infant’s physiology [306] including the immune system [307], temperature, and emotional state. Similarly, today’s music listeners use music to regulate their emotional state [282].

Geborgenheit corresponds to the second level in Maslow’s [308] hierarchy of psychological needs: 1 physiological needs, 2 safety and security, 3 social needs (love, belonging, affection, intimacy), 4 esteem, 5 self-actualization. It is the emotion of basic trust [309] or *Urvertrauen* [310]—the feeling that nothing serious can go wrong because one is in good hands. It gives infants the freedom to play and explore, to be creative and humorous. 

### 8.3. Predictions

Awe and geborgenheit are two categories of emotional experience that occur more often in ritual than in everyday life.
The sublime does not urgently press, from an existential point of view; it is nonsocial and noninteractive. Nevertheless, the perception of existential safety is crucial, especially for the natural sublime: Niagara and Denali are immense, of extraordinary beauty, powerful and moody beyond measure, but the experiencing person is—although very close by—safe.[291] (p. 124)

These two emotions are therefore relevant for a non-circular theory of the origin of religion and music. An MS-based theory makes the following predictions:Postnatal activation of MS produces otherworldly feelings (awe).Since an infant with its mother is objectively safe (protected) and warm (thermoregulated), postnatal MS activation evokes geborgenheit.Both awe and geborgenheit may be “experienced” when children and adults participate in rituals.Ritual participants cannot know where these special feelings come from, because their ontogeny precedes reflective consciousness and language.The best survival strategy for infants and children is to understand and follow maternal intentions. Rituals, therefore, often feature the additional emotion of *devotion* (to the group or its leader/s).Since the mother’s traditional role includes consoling an infant that experiences hunger, thirst, pain, anger, or distress, rituals (including music and religion) provide similar comfort.Since oxytocin relieves childbirth pain [311,312], and is shared by mother and fetus/infant, religious and musical activities also reduce pain and anxiety [313,314].

## 9. Evaluating the Evidence 

Evidence for the theory is plentiful and diverse, but seldom direct or convincing in isolation. The evidence might be compelling if the observations that are consistent with theory clearly outnumbered the observations that are not (quasi-objective criterion), or were considered more important than them (subjective criterion) (cf. [315]).

### 9.1. The Mother Schema

The psychological reality of MS is difficult to investigate directly. If every fetus and infant has MS, then it is not possible to compare an experimental group of fetuses with MS with a control group without it. If a fetus or infant has a sensory or cognitive deficit, MS will be correspondingly impoverished, but an empirical investigation of such an effect can hardly avoid confounds. Future research on the neurocognitive foundations of MS may enable non-invasive empirical investigation. Hypotheses about the relationship between brain development and MS could be tested, drawing on the neuroscience of human relationships [316]. But activation of the MS may be indistinguishable from activation of other social cognitive networks [317]. 

### 9.2. Skill Development 

The theory is consistent with developmental research on skills and abilities. Infant musical abilities are comparable in many respects with those of adult non-musicians [318,319]. More securely attached toddlers and children have superior linguistic abilities [320,321,322]. The linguistic ability of women may be slightly superior to that of men [323,324].

MS theory predicts that children are sensitive to both music [318] and religion [24] and that music enhances memory [325]. Children of mute mothers are predicted to be less religious or musical—but religiosity and musicality also depend on other factors. Children of mute mothers might turn to religion or music to make up for a resulting emotional lack—an example of compensation in religious behavior [326]. Similarly, it would be difficult to test whether the children of bed-ridden mothers are less sensitive to rhythm and dance. Religious and musical behaviors may depend more on cultural traditions than on direct prenatal experience, making direct empirical testing difficult.

### 9.3. Strong Musical Experiences

Much musical emotion is ineffable [327,328,329], consistent with a prelinguistic origin. Consider the following statements by Gabrielsson’s [43] respondents. The text in italics refers to accompanying speculative explanations in Table 1; Table 2 considers more general relationships between musical features and strong experiences.
It was as if the music had *blown away all my thoughts*. I no longer had any worries, but nor did I feel any joy. The only thing that filled me was a magnificent feeling.(p. 19)
These experiences feel the strongest; I *flight* off, go into *another state*, things around me change, there is *magic*, I *float* around in *some space or other*. I *go inwards* at the same time that I expand. All feelings are *mixed together*. (p. 27)
The music started, loud and magnificent. Suddenly I thought that *two strong arms lifted me* and put me in front of the dance group. All my fear disappeared. The music seized all my attention and the dance steps matched with the music and vice versa. A feeling of joy and happiness swept through me. *I was in a completely different world*. It was so fantastic that music and dance *fused together to one unit*.(p. 32)
Quietly, gently as a warm summer breeze, the music reached me and went right in. I sat there without moving a muscle, *leaning forward*, and just sort of *swallowed* it. I had never heard anything so *indescribably beautiful*. And *every bit of me was filled with divine harmony* and I was aware that tears were starting to run down my cheeks, but I didn’t care about that and I wasn’t ashamed.(p. 48)
It was a total experienced just as strong as *first love*. … How I gradually *lost contact with the ground* and experienced an *intoxication of the senses*. (p. 51)
Everything was calm and peaceful. I turned the radio on and out poured classical music from a big orchestra. I felt how I sucked in the music, *I drank it, swallowed it, was being filled by it. Like a dry sponge which is reacquiring its shape*. (p. 52)
It felt as if my legs were *filled with a fizzy drink* and I was lifted up to a *higher sphere*. If there is a *heaven* after this life, I have already visited it. My experience is similar to that which *near-death* patients describe: a journey towards light and happiness.(p. 53)
We sang together in such harmony that I’ve never experienced anything like it before. It felt as if I was *hovering and was lifted higher and higher up on the wings of music*! All my *problems disappeared* and my entire body with *dissolved in tones*. When the record came to an end it felt as if I had slept for 24 hours solid. *I felt so rested!*
(p. 59)
Then something amazing happened. I played as if I was in a *trance*, and I promised I have never played better. This was an incredible feeling, it was *as if time and space disappeared.* There was only me and the music, *nothing else existed*.(p. 63)
Suddenly I was in some inexplicable way *sucked into* the music. It felt as if I was somehow lifted up from my seat and sort of *floated in the room* while at the same time being filled with serenity and *inner harmony, in raptures*. (p. 79)
It seemed as though I was sitting in an enormous *globe of light*, filled with music. *All the earthly things around me had completely vanished*. The rustling of paper, the coughs and scrapings were gone. I was alone with the music, filled with joy, *unaware of everything else around me*. (p. 81)

Gabrielsson [43] documented statements by seven additional respondents who independently reported a feeling of weightless or floating (his Section 8.1) and nine who reported out-of-body experiences (Section 8.2).

### 9.4. Allusions to Mother Schema in the Christian Bible

A thorough evaluation should consider evidence of diverse kinds from diverse sources. The following well-known excerpts from Chapter 1 of the Book of John in the New Testament (New International Version), may be relevant:
1.In the beginning was the Word, and the Word was with God, and the Word was God.3.Through him all things were made; without him nothing was made that has been made.4.In him was life, and that life was the light of all mankind.5.The light shines in the darkness, and the darkness has not overcome it.12.Yet to all who did receive him, to those who believed in his name, he gave the right to become children of God.14.The Word became flesh and made his dwelling among us. 18.No one has ever seen God.


These counterintuitive statements “ring true” for Christians. Consider the following speculations:
1.The most important sound for the fetus is the mother’s voice [232]. In Christian theology, “Logos” (the word) has diverse meanings including Jesus as God, the soul of Jesus, a label for God, and God’s potentiality [332].3.From the fetal viewpoint, the mother is the universe. For the infant, she is the origin of life. 4–5.The shining light alludes to the sudden brightness of the postnatal world in which the infant “meets” the mother. 12.“Children of God” is consistent with mother-infant attachment. Many believe that supernatural agents watch over them, care about moral behavior, and punish wrong behavior [333]—as mothers do.14.“The Word became flesh” suggests accommodation (Piaget) of MS at birth, after which the infant identifies the mother not only by her voice (word) but also by vision, touch, and (postnatal) smell/taste.18.Fetal eyes respond to lightness, but there is nothing to see.


## 10. Epistemological Biases

Theories of the origin of religion and music depend on the author’s social, historic, cultural and political context (cf. [334]). The present approach is biased toward ecological psychology, feminism, and subjectivity. These biases may be related to each other (e.g., feminism and ecological theory [335]). 

### 10.1. Ecological Psychology

An ecological approach to human behavior (including fetal behavior) focuses on real objects and interactions in the physical world rather than abstract cognitive processes. It considers affordances: how an organism can use objects and interactions and what they offer the organism [336,337]. These ideas can be applied to the fetus: fetal perception, like any other perception, is active, and fetal affordances are relevant for (postnatal) survival. Wilson [338] argued that environmental interactions are more directly observable and accessible to scientific method (data analysis, model testing) than inner cognitive processes:
(1) cognition is situated; (2) cognition is time-pressured; (3) we off-load cognitive work onto the environment; (4) the environment is part of the cognitive system; (5) cognition is for action; (6) offline cognition is body based.(p. 625)

Stewart and colleagues [339] argued that “cognition is grounded in the sensorimotor dynamics of the interactions between a living organism and its environment” (p. vii). In an ecological, embodied, enactivist paradigm, consciousness (everything of which humans are conscious, including supernatural agents) is based on embodied interactions with the human physical and social environment [340]. 

The words “mind” and “mental” can be misleading. Rossano [170] listed the following “mental attributes” of human religious beliefs: “one, agency detection and causal attribution; two, the social and emotional commitments of group living; three, narrative formation and the emergence of existential anxieties; and four, the ecstatic or mystical experience”. These religious behaviors and experiences may be better regarded as observable forms of interaction between humans and their social and physical environment than as abstract cognitive structures, representations, or modules.

An ecological approach is skeptical of neurophysiological accounts of behavioral origins such as the following:
the amygdala, hippocampus, and temporal lobe … are responsible for religious, spiritual, and mystical trancelike states, dreaming, astral projection, near-death and out-of-body experiences, and the hallucination of ghosts, demons, angels, and gods.[341] (p. 105)

While the phenomena in question would be impossible without the mentioned neurophysiological substrates, these substrates are unlikely to represent the ultimate causes of the behavior.

### 10.2. Feminism

The theory is feminist in the sense that modern human behavior is held to be based primarily on women’s roles and behaviors in ancient hunter-gatherer societies. In this way, the theory responds to accusations of sexism in evolutionary theory [342]. Male-oriented theories focus on hunting and fighting [10] rather than childcare and gathering [343]. In 1960s popular anthropology, “men were making and using tools, obtaining and sharing food, forming male-male alliances, developing communication, and becoming intelligent for more effective hunting”, although women “must have been mobile, obtaining food, sharing, carrying offspring and playing a central role in socialization of the young and in social interaction” [344] (p. 12). 

Today’s world religions are dominated by male leaders, but women have always dominated early moral training [345]. Whereas carers and infants are not equals, which makes carer-infant interactions a poor model for adult interactions, carer-infant relationships are nevertheless founded on mutual empathy [346]—an ability that is crucial for both carers and infants if the infant is to survive in dangerous situations. In this way, carer-infant interactions can parsimoniously explain aspects of
the evolved taste for “fairness” in dyadic and small-scale interpersonal interaction [which] appears to be directly related to evolved neural mechanisms that are also are activated in the application of punishment by impartial judges in modern legal systems. [8] (p. 192)

MS theory suggests that infants and children themselves contributed significantly and actively to the origin of religion and music. Trevathan and Rosenberg [74] commented that
Many of the distinctive characteristics that make us human can trace their origins (or at least their significance) to the fact that we give birth to infants who are highly dependent on others … While we recognize that single-cause explanations of the human adaptation are simplistic, we propose that an equally important player in the story of human evolution … is the helpless, attractive human infant.(pp. 1–2)

In other respects, the present theory is not feminist. For example, it does not directly involve any of the following four points:
Practically, the process of feminist research is characterized by four primary features: (1) expanding methodologies to include both quantitative and qualitative methods, (2) connecting women for group-level data collection, (3) reducing the hierarchical relationship between researchers and their participants to facilitate trust and disclosure, and (4) recognizing and reflecting upon the emotionality of women’s lives.[347] (p. 773)

### 10.3. Subjectivity 

Like music, religion involves special, mystical experiences and emotions [179,348]. For a skeptical scientist or philosopher, accounts of supernatural encounters may be considered evidence of god’s existence within the world of subjective human experience, but not in the physical world. That interpretation raises interesting issues for the philosophy of reality and the psychology of operationalization, and motivates the search for plausible scientific explanations. 

Whereas philosophical materialists claim that subjective experiences can be reduced to brain states, the mind-body problem remains unsolved and may be intrinsically unsolvable [349]. Dualist philosophies of mind acknowledge the existence of subjectivity/qualia as fundamentally different from, but intimately connected to, the objective physics/physiology of the human brain, body, and environment (e.g., [350]). Relevant interpretations include property dualism, substance dualism, emergent materialism, and subjective physicalism (cf. [351]).

A possible solution is *epiphenomenalism* (perhaps in combination with *evolutionism* [352]), according to which subjective states may be caused by physical states but not vice-versa. While the physical world functions independently of consciousness, subjective reality is only partly independent of physical reality. 

The present study takes an epiphenomenal middle path between the extremes of philosophical materialism and mind-body dualism. A quantitative-empirical psychological approach is supplemented by experiential reports from qualitative interview studies. The divine/spiritual presence experienced by ritual participants across cultures is considered psychologically real, even in the absence of corresponding external physical reality.

## 11. Conclusions

The idea that all aspects of human behavioral modernity may have a single cause—the obstetric dilemma—is intriguing and promising, but also problematic. Human behavior is complex and the evidence (archeological, historical, sociological, theological, music-theoretical, acoustical, physiological, neurological, psychological, philosophical) is circumstantial. Given the many theories and considerations that might contribute to an explanation of behavioral modernity and its emergence, a theory of this kind can hardly be “falsifiable” or “evidence-based” (cf. [353,354,355]). Instead, new theories tend to be accepted when intolerable anomalies are found in existing theories and evidence for the new is considered subjectively by experts to exceed evidence for the old [315]. Besides, even in a relatively positive or optimistic evaluation, the present theory can hardly stand alone; it tends to complement other theories of the origin of language, religion, and music, rather than undermining them.

It is nevertheless interesting to consider the following “hard thesis”: If there had not been an obstetric dilemma, and if as a consequence human mothers and infants had not developed new forms of communication, modern human behavior including language, religion and music, and hence humanity as we know it, would not have emerged. I have tentatively addressed several of the issues thrown up by this radical hypothesis. The project is highly interdisciplinary and a thorough evaluation should ideally be carried out in all relevant disciplines.

## Figures and Tables

**Figure 1 behavsci-09-00142-f001:**
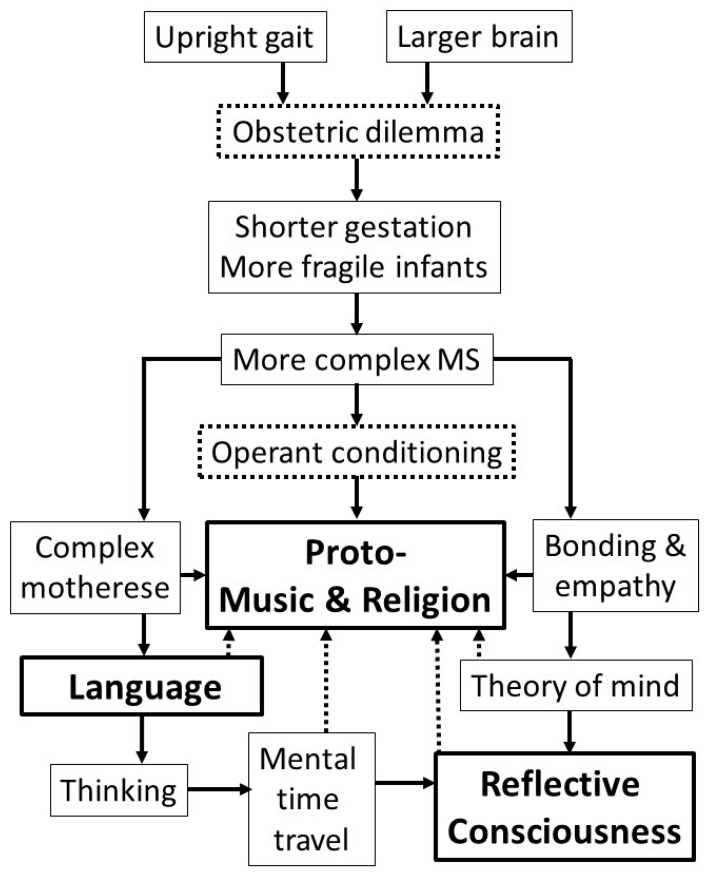
Sketch of the structure of the main thesis. The main behavioral phenomena to be explained are bold. Assumed primary causal relationships are illustrated as arrows with full lines. Dotted arrows denote secondary or circular causalities. Dotted boxes denote processes rather than outcomes.

**Table 1 behavsci-09-00142-t001:** Speculative Explanations of Data from Gabrielsson [43] (see text).

Recurring Elements in Reports of Strong Musical Experiences	Explanations Based on MS Theory
Disappearance of thoughts and problems, a different state of consciousness, trance, intoxication of the senses, no sense of time, indescribable	The fetus/infant is prelinguistic and lacks reflective consciousness including a sense of the future and past relative to the present.
Floating, hovering, weightlessness, flying, no contact with ground, out-of-body experience, liquid, water, waves; drinking, swallowing, sucking; pouring, dissolved	The fetus is floating in amniotic fluid. It is protected by it and sometimes swallows it (cf. [120]). Christian baptism, Jewish Tvilah, Islamic Wuḍū, and Shinto Misogi (ritual purification) may be relevant.
Another world, no sense of place, nothing else exists, heaven, near-death experience	The uterine environment is closed and separate from the rest of the world.
Globe of light	The eye develops throughout the prenatal period [330]; the human fetus may perceive light that penetrates the uterus [331].
God on high, being in heaven	The mother’s vocal chords and heart are physically higher than the fetus. (This explanation is problematic because it depends on both a functioning vestibular system and directional hearing in utero.)
Sense of fusion, everything mixed together, harmony, every bit of me filled	The fetus is part of (and inseparable from) the mother.
Attention completely focused on music or divine encounter; absorption, unaware of all else	This may be a figure-ground effect as the fetus perceives the mother as separate from background noise.
Emotions in the categories transcendence and geborgenheit, raptures, indescribably beautiful	These may be adult attempts to retrospectively describe emotions of the fetus and infant when perceiving the mother.
Magic, enchantment	The fetus and newborn lack concepts of cause and effect.
Recovery	Attachment to the mother ensures safety and enables recovery [62].
Love (“first love”)	The fetus/infant-mother relationship is literally the first love in the lifespan.

**Table 2 behavsci-09-00142-t002:** Musical Features that Evoke Strong Experiences.

Situations That Trigger Strong Religious or Music Experiences	Speculative Explanations Based on MS Theory
Musical melody	Sounds like mother’s voice through filter of amniotic fluid (low-pass filter removes high frequencies and shifts attention from timbral changes to frequency contours)
Musical rhythm and movement	Sounds like mother’s heartbeat or footsteps; feels like maternal walk
Musical harmony	Sounds like the lower harmonics of the mother’s voice (spectral pitch pattern) before maturation of harmonic pitch pattern recognition (virtual pitch)
Sound energy mainly at lower frequencies (100-300 Hz), e.g. highly resonant rooms or caves	Amniotic fluid muffles high frequencies
Bent posture (kneeling, bowing, hands touching each other and face etc.)	Fetal positions
Repetition (with variation) of sound or movement patterns [91,92]	Prenatally perceptible sound and movement patterns are repeated many times with variation

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
