# Peer review of "Mother Schema, Obstetric Dilemma, and the Origin of Behavioral Modernity"

_behavsci, 2019, doi:10.3390/bs9120142_

Round 1

Reviewer 1 Report

This is a theoretical article that synthesizes a large number of fields to address a Large Subject. The author shows broad and deep learning. As he says, his hypothesis cannot be falsified; the article is of value, nevertheless, because it competently and coherently draws together supportive knowledge from a variety of subject areas. It is a stimulating paper. On the review of the article I have made a number of suggestions, which the author may or may not wish to follow.

Author Response

I carefully considered every point made by Reviewer 1 and changed the text accordingly. The editor asked me to "consider critical revisions (or rebuttals) that attend to Reviewer 2 comments" implying that a detailed response to Reviewer 1 was not necessary. I will be glad to provide a detailed response to Reviewer 1 if asked. That would involve copying the comments that Reviewer 1 inserted into the pdf submission into a separate file and documenting the corresponding changes.

Reviewer 2 Report

The subject of this paper is timely. There has been considerable work about the mother’s infant schema (the “cute schema”, but since Konrad Lorenz’ early work on imprinting of goslings there has been virtually no interest in the converse, the infant’s mother schema. The main importance, and that could be stated more clearly in this paper, is that both the mother schema and the mother’s infant schema are the source of adult attachment and love. Presumably, the same brain mechanisms that are used for the mother-infant relationship are also used for adult relationships, as has been shown in the case of oxytocin which obviously acquired its role in social bonding initially in the form of triggering maternal behavior. Also worth mentioning is that all mammals are supposed to have both the mother’s infant schema and the infant’s mother schema, but that only social species have co-opted the brain mechanisms for social bonding among adults and only a few including the monogamous ones) are using them for bonding between mates.

There is also the question of the conceptual separation between the schemas, which are quite specific sensory-cognitive structures dependent on sensory input, and the emotional “resources” that are activated by the schemas. In drug-induced ecstasy, for example, or in music, emotions are triggered that are normally triggered by the mechanisms of social bonding. We can say that these emotional resources exist for the purpose of being triggered by mother schema and related schemata, but the emotion as such is different from the schema. Awe, for example, can be triggered by the mother schema, but also by interactions with a powerful alpha male. Presumably, the same emotional circuits but different triggers.

Some specific points:

Line 32: I wouldn't identify Cromagnon with Homo sapiens sapiens. The latter is the branch of H. sapiens that evolved in Africa at the time of the Neanderthals and Denisovans, and finally took over the world. The term Cromagnon should be limited to the earliest modern human race in Europe.

Line 54: Perhaps behavioral modernity is not such a great mystery at all, if there has been selection for higher intelligence for a long time. Intelligence would be considered a general-purpose mechanism that allows abstract concepts, including "mental time travel" to imagine future states of the world and the consequences of one's present and intended actions. Being of general usefulness, intelligence is responsible for a whole bunch of behaviors that archaeologists describe as “behavioral modernity”, as well as for those that historians describe as “civilization”.

Line 70-71: We are not really dealing here with "fetal cognitive abilities", but with cognitive abilities that start developing in the fetus but that reach a fully functional state of maturity only when the child starts talking. Linguistic ability is based on simple associative learning: a predisposition to associate sound patterns with things that are perceived through the senses, and the ability and motivation to reproduce these perceived sound patterns through the vocal apparatus. Later on, associations are not only with perceived things but also with abstract concepts that are in turn associated with "real" things that can be perceived through the senses.

Line 73: Here you will have to explain why a “prenatal mother schema” should exist in the first place. I cannot see any evolutionary advantage in having it. After birth it is important for the infant to recognize the mother and behave in a way that encourages continued maternal care and prevents infanticide. However, there is no such selective pressure before birth.

Line 85ff: Here you seem to assume that the mother schema develops in the fetus and infant and then persists as such in the adult, buried somehow in the depths of the mind but sometimes triggered by religious/ritualistic kinds of stimuli. Perhaps a more productive way of looking at it is that during ontogenesis the mother schema is transformed into something else, something more general. It seems to be obvious that adult forms of love encompass caring (derived both ontogenetically and phylogenetically from maternal care) and attachment (derived in large part from the infant's mother schema. In a parallel strand of ontogenetic development, the infant's more general experience of all-powerful adults develops into the perception of power hierarchies. It should be obvious that these two parallel ontogenetic paths, from mother schema to adult attachments and from the experience of infant dependence on powerful adults to dominance hierarchies, are the foundations of religious experience. And, it should be emphasized, the entirety of human sociality.

Line 91-92: Why should there be prenatal episodic memory? It makes no evolutionary sense. Memory is for learning, and learning makes sense only when it can influence future behaviour in adaptive ways. The fetus cannot manipulate the mother's behaviour through cognitive means, and things learned during fetal life would be useless after birth when the survival challenges are entirely different. If there is no survival advantage to prenatal memory, either it didn’t evolve at all or if it did, only as a byproduct of something that is indeed adaptive. The latter could only be postnatal episodic memory, but humans are born so prematurely that it is even doubtful that infants in the first year after birth have much of it.

Line 101ff: Somewhere you will have to explain what exactly "reflective consciousness" is, and what it isn't.

Line 130-132: Again, I doubt that things learned before birth can in any way be useful after birth. Also, being born at a more immature stage, as is the case in humans compared to other primates, implies lower, not higher ability to learn during the first months after birth.

Line 166ff: Suggestion: For religion, you may consider a conceptual distinction between two sources of religion. The first, which is directly derived from mother schema and social dominance, is the idea of very powerful sentient beings (gods) that may or may not interfere with human affairs. The second is derived from the contemplation of man's place in the world and especially his mortality, meaning it requires a higher stage of cognitive development. This second element produced the concept of mind-body dualism, as in the brahman-atman equivalence of the Vedanta and ideas of an afterlife and/or reincarnation. There may be different evolutionary, neurological and ontogenetic sources of these two kinds of "religious" ideas.

Line 224-225: Obviously, "complex reflective language" did not emerge suddenly. It must have evolved from a protolanguage that was based on nothing more than the ability, and the inclination, to associate complex sound patterns with things in the real world. This ability and inclination could evolve only in the presence of an evolutionary advantage. This evolutionary advantage was given only when human ancestors had acquired the ability to produce these complex sounds. It has been speculated that anatomical changes in the vocal apparatus associated with upright walking enabled speech. If so, the evolution of a protolanguage may date back several million years and may have driven brain evolution from Australopithecus to early Homo.

Line 230-231: The human-chimpanzee split happened perhaps a bit earlier, because mutation rates are lower than originally assumed, and therefore the evolutionary clock had to be recalibrated.

Line 323-314: Absolutely dispensable, or absolutely indispensable?

Line 346ff: Here, it seems, the argument is made that the mother-infant relationship is the source of selective pressure for the evolution of (proto)language. Let's not forget that language develops only at the age of about 2 years, while mortality is highest in the first year after birth. Motherese looks more like a hybrid of language and emotional expression, used at a time when baby’s linguistic development is still rudimentary, although it could be argued that protolanguage (though not more complex, grammatical language) evolved in part for mother-infant interactions and teaching.

Line 391-392: Music and religion arose in the context of ritual? This would suggest that ritual came before religion and music. As far as I know, ritual is a relative latecomer in cultural evolution, being strongly related to cultural complexity. See Zern, D. S. (1984). Religiousness related to cultural complexity and pressures to obey cultural norms. Genetic psychology monographs 110: 207-227. So, more likely religion and music were there for some time already when people started with rituals.

Line 458ff: Most of these postures have not much to do with the fetal position but are part of the repertoire of behaviors related to social dominance: making oneself smaller and defenseless by bowing, kneeling, prostration etc. These are gestures of submission to an alpha male, not re-enactments of the fetal position. They are used in interactions with a personal god, whereas a more upright position in meditation is related not to appeasement of a personal god but to an approach to the unknowable mysteries of life and existence. You may be more on target with dance, because being carried by the mother has a strong soothing effect on the infant, and such preference for rhythmic movement may persist into adulthood when it is expressed as dance.

Line 469ff: Here, I think, you should better work out the ontogenetic changes. There is infant attachment to the mother. In adult life, people depend no longer on the mother but they do depend on the groups to which they belong. This certainly was the case for prehistoric hunter-gatherer bands. What plausibly happens is that the neurological systems for infant attachment to the mother are developed into a new function: supporting the individual’s attachment to family, country, or any other group with which he identifies. Mother attachment as the evolutionary and ontogenetic source of positive (but not negative) ethnocentrism.

Line 541ff: Here we are dealing less with an activation of the mother schema as such, but with an activation of the pool of emotional resources that ordinarily become activated by the mother schema in infants and by related stimuli (friends, family members, or more abstract group identities) in adulthood.

Line 847-848: Transnatal memory is predicted to be not robust at all because experiences before birth are unlikely to lead to appropriate responses after birth, which are, during the first months at least, almost purely reflexive anyhow. Again, memory is learning, and learning makes sense only when the conditions at learning are maintained long enough for appropriate behavioural responses at a later time. When the conditions change as radically as they do at birth, any prenatal learning will be useless after birth.

Line 909: Oxytocin is an interesting case because empirical studies show its involvement in adult bonding although its primary function is in parturition and lactation. It shows that the evolutionary and ontogenetic source of adult bonding lies in the mother-infant relationship, since one of the original functions of oxytocin appears to be to activate the mother's care for the infant after birth. You may point out that so far we know of no biochemical marker for the mother schema, no chemical we know of that specifically triggers infant attachment to the mother, but it is perfectly plausible that the infant's mother schema carries over into adult life the way the mother's infant schema carries over into adult life, that indeed the two merge during ontogenesis to form stable adult attachments. Dopamine and opioids appear to be associated with general emotional resources that can be described as positive-euphoriant and tranquilizing (repressing negative emotions), respectively.

Line 1129ff: This section is really very speculative. The connection with the experience of birth is quite improbable.

Line 1186ff: One way in which the theory is not feminist at all is in that it starts with the mother schema, and then claims that attachment to God is equivalent to the infant's attachment to the mother. In that case, wouldn't monotheistic religions imagine their god as female? I only know God as a bearded old man.

The subject of this paper is timely. There has been considerable work about the mother’s infant schema (the “cute schema”, but since Konrad Lorenz’ early work on imprinting of goslings there has been virtually no interest in the converse, the infant’s mother schema. The main importance, and that could be stated more clearly in this paper, is that both the mother schema and the mother’s infant schema are the source of adult attachment and love. Presumably, the same brain mechanisms that are used for the mother-infant relationship are also used for adult relationships, as has been shown in the case of oxytocin which obviously acquired its role in social bonding initially in the form of triggering maternal behavior. Also worth mentioning is that all mammals are supposed to have both the mother’s infant schema and the infant’s mother schema, but that only social species have co-opted the brain mechanisms for social bonding among adults and only a few including the monogamous ones) are using them for bonding between mates.

There is also the question of the conceptual separation between the schemas, which are quite specific sensory-cognitive structures dependent on sensory input, and the emotional “resources” that are activated by the schemas. In drug-induced ecstasy, for example, or in music, emotions are triggered that are normally triggered by the mechanisms of social bonding. We can say that these emotional resources exist for the purpose of being triggered by mother schema and related schemata, but the emotion as such is different from the schema. Awe, for example, can be triggered by the mother schema, but also by interactions with a powerful alpha male. Presumably, the same emotional circuits but different triggers.

Some specific points:

Line 32: I wouldn't identify Cromagnon with Homo sapiens sapiens. The latter is the branch of H. sapiens that evolved in Africa at the time of the Neanderthals and Denisovans, and finally took over the world. The term Cromagnon should be limited to the earliest modern human race in Europe.

Line 54: Perhaps behavioral modernity is not such a great mystery at all, if there has been selection for higher intelligence for a long time. Intelligence would be considered a general-purpose mechanism that allows abstract concepts, including "mental time travel" to imagine future states of the world and the consequences of one's present and intended actions. Being of general usefulness, intelligence is responsible for a whole bunch of behaviors that archaeologists describe as “behavioral modernity”, as well as for those that historians describe as “civilization”.

Line 70-71: We are not really dealing here with "fetal cognitive abilities", but with cognitive abilities that start developing in the fetus but that reach a fully functional state of maturity only when the child starts talking. Linguistic ability is based on simple associative learning: a predisposition to associate sound patterns with things that are perceived through the senses, and the ability and motivation to reproduce these perceived sound patterns through the vocal apparatus. Later on, associations are not only with perceived things but also with abstract concepts that are in turn associated with "real" things that can be perceived through the senses.

Line 73: Here you will have to explain why a “prenatal mother schema” should exist in the first place. I cannot see any evolutionary advantage in having it. After birth it is important for the infant to recognize the mother and behave in a way that encourages continued maternal care and prevents infanticide. However, there is no such selective pressure before birth.

Line 85ff: Here you seem to assume that the mother schema develops in the fetus and infant and then persists as such in the adult, buried somehow in the depths of the mind but sometimes triggered by religious/ritualistic kinds of stimuli. Perhaps a more productive way of looking at it is that during ontogenesis the mother schema is transformed into something else, something more general. It seems to be obvious that adult forms of love encompass caring (derived both ontogenetically and phylogenetically from maternal care) and attachment (derived in large part from the infant's mother schema. In a parallel strand of ontogenetic development, the infant's more general experience of all-powerful adults develops into the perception of power hierarchies. It should be obvious that these two parallel ontogenetic paths, from mother schema to adult attachments and from the experience of infant dependence on powerful adults to dominance hierarchies, are the foundations of religious experience. And, it should be emphasized, the entirety of human sociality.

Line 91-92: Why should there be prenatal episodic memory? It makes no evolutionary sense. Memory is for learning, and learning makes sense only when it can influence future behaviour in adaptive ways. The fetus cannot manipulate the mother's behaviour through cognitive means, and things learned during fetal life would be useless after birth when the survival challenges are entirely different. If there is no survival advantage to prenatal memory, either it didn’t evolve at all or if it did, only as a byproduct of something that is indeed adaptive. The latter could only be postnatal episodic memory, but humans are born so prematurely that it is even doubtful that infants in the first year after birth have much of it.

Line 101ff: Somewhere you will have to explain what exactly "reflective consciousness" is, and what it isn't.

Line 130-132: Again, I doubt that things learned before birth can in any way be useful after birth. Also, being born at a more immature stage, as is the case in humans compared to other primates, implies lower, not higher ability to learn during the first months after birth.

Line 166ff: Suggestion: For religion, you may consider a conceptual distinction between two sources of religion. The first, which is directly derived from mother schema and social dominance, is the idea of very powerful sentient beings (gods) that may or may not interfere with human affairs. The second is derived from the contemplation of man's place in the world and especially his mortality, meaning it requires a higher stage of cognitive development. This second element produced the concept of mind-body dualism, as in the brahman-atman equivalence of the Vedanta and ideas of an afterlife and/or reincarnation. There may be different evolutionary, neurological and ontogenetic sources of these two kinds of "religious" ideas.

Line 224-225: Obviously, "complex reflective language" did not emerge suddenly. It must have evolved from a protolanguage that was based on nothing more than the ability, and the inclination, to associate complex sound patterns with things in the real world. This ability and inclination could evolve only in the presence of an evolutionary advantage. This evolutionary advantage was given only when human ancestors had acquired the ability to produce these complex sounds. It has been speculated that anatomical changes in the vocal apparatus associated with upright walking enabled speech. If so, the evolution of a protolanguage may date back several million years and may have driven brain evolution from Australopithecus to early Homo.

Line 230-231: The human-chimpanzee split happened perhaps a bit earlier, because mutation rates are lower than originally assumed, and therefore the evolutionary clock had to be recalibrated.

Line 323-314: Absolutely dispensable, or absolutely indispensable?

Line 346ff: Here, it seems, the argument is made that the mother-infant relationship is the source of selective pressure for the evolution of (proto)language. Let's not forget that language develops only at the age of about 2 years, while mortality is highest in the first year after birth. Motherese looks more like a hybrid of language and emotional expression, used at a time when baby’s linguistic development is still rudimentary, although it could be argued that protolanguage (though not more complex, grammatical language) evolved in part for mother-infant interactions and teaching.

Line 391-392: Music and religion arose in the context of ritual? This would suggest that ritual came before religion and music. As far as I know, ritual is a relative latecomer in cultural evolution, being strongly related to cultural complexity. See Zern, D. S. (1984). Religiousness related to cultural complexity and pressures to obey cultural norms. Genetic psychology monographs 110: 207-227. So, more likely religion and music were there for some time already when people started with rituals.

Line 458ff: Most of these postures have not much to do with the fetal position but are part of the repertoire of behaviors related to social dominance: making oneself smaller and defenseless by bowing, kneeling, prostration etc. These are gestures of submission to an alpha male, not re-enactments of the fetal position. They are used in interactions with a personal god, whereas a more upright position in meditation is related not to appeasement of a personal god but to an approach to the unknowable mysteries of life and existence. You may be more on target with dance, because being carried by the mother has a strong soothing effect on the infant, and such preference for rhythmic movement may persist into adulthood when it is expressed as dance.

Line 469ff: Here, I think, you should better work out the ontogenetic changes. There is infant attachment to the mother. In adult life, people depend no longer on the mother but they do depend on the groups to which they belong. This certainly was the case for prehistoric hunter-gatherer bands. What plausibly happens is that the neurological systems for infant attachment to the mother are developed into a new function: supporting the individual’s attachment to family, country, or any other group with which he identifies. Mother attachment as the evolutionary and ontogenetic source of positive (but not negative) ethnocentrism.

Line 541ff: Here we are dealing less with an activation of the mother schema as such, but with an activation of the pool of emotional resources that ordinarily become activated by the mother schema in infants and by related stimuli (friends, family members, or more abstract group identities) in adulthood.

Line 847-848: Transnatal memory is predicted to be not robust at all because experiences before birth are unlikely to lead to appropriate responses after birth, which are, during the first months at least, almost purely reflexive anyhow. Again, memory is learning, and learning makes sense only when the conditions at learning are maintained long enough for appropriate behavioural responses at a later time. When the conditions change as radically as they do at birth, any prenatal learning will be useless after birth.

Line 909: Oxytocin is an interesting case because empirical studies show its involvement in adult bonding although its primary function is in parturition and lactation. It shows that the evolutionary and ontogenetic source of adult bonding lies in the mother-infant relationship, since one of the original functions of oxytocin appears to be to activate the mother's care for the infant after birth. You may point out that so far we know of no biochemical marker for the mother schema, no chemical we know of that specifically triggers infant attachment to the mother, but it is perfectly plausible that the infant's mother schema carries over into adult life the way the mother's infant schema carries over into adult life, that indeed the two merge during ontogenesis to form stable adult attachments. Dopamine and opioids appear to be associated with general emotional resources that can be described as positive-euphoriant and tranquilizing (repressing negative emotions), respectively.

Line 1129ff: This section is really very speculative. The connection with the experience of birth is quite improbable.

Line 1186ff: One way in which the theory is not feminist at all is in that it starts with the mother schema, and then claims that attachment to God is equivalent to the infant's attachment to the mother. In that case, wouldn't monotheistic religions imagine their god as female? I only know God as a bearded old man.

The subject of this paper is timely. There has been considerable work about the mother’s infant schema (the “cute schema”, but since Konrad Lorenz’ early work on imprinting of goslings there has been virtually no interest in the converse, the infant’s mother schema. The main importance, and that could be stated more clearly in this paper, is that both the mother schema and the mother’s infant schema are the source of adult attachment and love. Presumably, the same brain mechanisms that are used for the mother-infant relationship are also used for adult relationships, as has been shown in the case of oxytocin which obviously acquired its role in social bonding initially in the form of triggering maternal behavior. Also worth mentioning is that all mammals are supposed to have both the mother’s infant schema and the infant’s mother schema, but that only social species have co-opted the brain mechanisms for social bonding among adults and only a few including the monogamous ones) are using them for bonding between mates.

There is also the question of the conceptual separation between the schemas, which are quite specific sensory-cognitive structures dependent on sensory input, and the emotional “resources” that are activated by the schemas. In drug-induced ecstasy, for example, or in music, emotions are triggered that are normally triggered by the mechanisms of social bonding. We can say that these emotional resources exist for the purpose of being triggered by mother schema and related schemata, but the emotion as such is different from the schema. Awe, for example, can be triggered by the mother schema, but also by interactions with a powerful alpha male. Presumably, the same emotional circuits but different triggers.

Some specific points:

Line 32: I wouldn't identify Cromagnon with Homo sapiens sapiens. The latter is the branch of H. sapiens that evolved in Africa at the time of the Neanderthals and Denisovans, and finally took over the world. The term Cromagnon should be limited to the earliest modern human race in Europe.

Line 54: Perhaps behavioral modernity is not such a great mystery at all, if there has been selection for higher intelligence for a long time. Intelligence would be considered a general-purpose mechanism that allows abstract concepts, including "mental time travel" to imagine future states of the world and the consequences of one's present and intended actions. Being of general usefulness, intelligence is responsible for a whole bunch of behaviors that archaeologists describe as “behavioral modernity”, as well as for those that historians describe as “civilization”.

Line 70-71: We are not really dealing here with "fetal cognitive abilities", but with cognitive abilities that start developing in the fetus but that reach a fully functional state of maturity only when the child starts talking. Linguistic ability is based on simple associative learning: a predisposition to associate sound patterns with things that are perceived through the senses, and the ability and motivation to reproduce these perceived sound patterns through the vocal apparatus. Later on, associations are not only with perceived things but also with abstract concepts that are in turn associated with "real" things that can be perceived through the senses.

Line 73: Here you will have to explain why a “prenatal mother schema” should exist in the first place. I cannot see any evolutionary advantage in having it. After birth it is important for the infant to recognize the mother and behave in a way that encourages continued maternal care and prevents infanticide. However, there is no such selective pressure before birth.

Line 85ff: Here you seem to assume that the mother schema develops in the fetus and infant and then persists as such in the adult, buried somehow in the depths of the mind but sometimes triggered by religious/ritualistic kinds of stimuli. Perhaps a more productive way of looking at it is that during ontogenesis the mother schema is transformed into something else, something more general. It seems to be obvious that adult forms of love encompass caring (derived both ontogenetically and phylogenetically from maternal care) and attachment (derived in large part from the infant's mother schema. In a parallel strand of ontogenetic development, the infant's more general experience of all-powerful adults develops into the perception of power hierarchies. It should be obvious that these two parallel ontogenetic paths, from mother schema to adult attachments and from the experience of infant dependence on powerful adults to dominance hierarchies, are the foundations of religious experience. And, it should be emphasized, the entirety of human sociality.

Line 91-92: Why should there be prenatal episodic memory? It makes no evolutionary sense. Memory is for learning, and learning makes sense only when it can influence future behaviour in adaptive ways. The fetus cannot manipulate the mother's behaviour through cognitive means, and things learned during fetal life would be useless after birth when the survival challenges are entirely different. If there is no survival advantage to prenatal memory, either it didn’t evolve at all or if it did, only as a byproduct of something that is indeed adaptive. The latter could only be postnatal episodic memory, but humans are born so prematurely that it is even doubtful that infants in the first year after birth have much of it.

Line 101ff: Somewhere you will have to explain what exactly "reflective consciousness" is, and what it isn't.

Line 130-132: Again, I doubt that things learned before birth can in any way be useful after birth. Also, being born at a more immature stage, as is the case in humans compared to other primates, implies lower, not higher ability to learn during the first months after birth.

Line 166ff: Suggestion: For religion, you may consider a conceptual distinction between two sources of religion. The first, which is directly derived from mother schema and social dominance, is the idea of very powerful sentient beings (gods) that may or may not interfere with human affairs. The second is derived from the contemplation of man's place in the world and especially his mortality, meaning it requires a higher stage of cognitive development. This second element produced the concept of mind-body dualism, as in the brahman-atman equivalence of the Vedanta and ideas of an afterlife and/or reincarnation. There may be different evolutionary, neurological and ontogenetic sources of these two kinds of "religious" ideas.

Line 224-225: Obviously, "complex reflective language" did not emerge suddenly. It must have evolved from a protolanguage that was based on nothing more than the ability, and the inclination, to associate complex sound patterns with things in the real world. This ability and inclination could evolve only in the presence of an evolutionary advantage. This evolutionary advantage was given only when human ancestors had acquired the ability to produce these complex sounds. It has been speculated that anatomical changes in the vocal apparatus associated with upright walking enabled speech. If so, the evolution of a protolanguage may date back several million years and may have driven brain evolution from Australopithecus to early Homo.

Line 230-231: The human-chimpanzee split happened perhaps a bit earlier, because mutation rates are lower than originally assumed, and therefore the evolutionary clock had to be recalibrated.

Line 323-314: Absolutely dispensable, or absolutely indispensable?

Line 346ff: Here, it seems, the argument is made that the mother-infant relationship is the source of selective pressure for the evolution of (proto)language. Let's not forget that language develops only at the age of about 2 years, while mortality is highest in the first year after birth. Motherese looks more like a hybrid of language and emotional expression, used at a time when baby’s linguistic development is still rudimentary, although it could be argued that protolanguage (though not more complex, grammatical language) evolved in part for mother-infant interactions and teaching.

Line 391-392: Music and religion arose in the context of ritual? This would suggest that ritual came before religion and music. As far as I know, ritual is a relative latecomer in cultural evolution, being strongly related to cultural complexity. See Zern, D. S. (1984). Religiousness related to cultural complexity and pressures to obey cultural norms. Genetic psychology monographs 110: 207-227. So, more likely religion and music were there for some time already when people started with rituals.

Line 458ff: Most of these postures have not much to do with the fetal position but are part of the repertoire of behaviors related to social dominance: making oneself smaller and defenseless by bowing, kneeling, prostration etc. These are gestures of submission to an alpha male, not re-enactments of the fetal position. They are used in interactions with a personal god, whereas a more upright position in meditation is related not to appeasement of a personal god but to an approach to the unknowable mysteries of life and existence. You may be more on target with dance, because being carried by the mother has a strong soothing effect on the infant, and such preference for rhythmic movement may persist into adulthood when it is expressed as dance.

Line 469ff: Here, I think, you should better work out the ontogenetic changes. There is infant attachment to the mother. In adult life, people depend no longer on the mother but they do depend on the groups to which they belong. This certainly was the case for prehistoric hunter-gatherer bands. What plausibly happens is that the neurological systems for infant attachment to the mother are developed into a new function: supporting the individual’s attachment to family, country, or any other group with which he identifies. Mother attachment as the evolutionary and ontogenetic source of positive (but not negative) ethnocentrism.

Line 541ff: Here we are dealing less with an activation of the mother schema as such, but with an activation of the pool of emotional resources that ordinarily become activated by the mother schema in infants and by related stimuli (friends, family members, or more abstract group identities) in adulthood.

Line 847-848: Transnatal memory is predicted to be not robust at all because experiences before birth are unlikely to lead to appropriate responses after birth, which are, during the first months at least, almost purely reflexive anyhow. Again, memory is learning, and learning makes sense only when the conditions at learning are maintained long enough for appropriate behavioural responses at a later time. When the conditions change as radically as they do at birth, any prenatal learning will be useless after birth.

Line 909: Oxytocin is an interesting case because empirical studies show its involvement in adult bonding although its primary function is in parturition and lactation. It shows that the evolutionary and ontogenetic source of adult bonding lies in the mother-infant relationship, since one of the original functions of oxytocin appears to be to activate the mother's care for the infant after birth. You may point out that so far we know of no biochemical marker for the mother schema, no chemical we know of that specifically triggers infant attachment to the mother, but it is perfectly plausible that the infant's mother schema carries over into adult life the way the mother's infant schema carries over into adult life, that indeed the two merge during ontogenesis to form stable adult attachments. Dopamine and opioids appear to be associated with general emotional resources that can be described as positive-euphoriant and tranquilizing (repressing negative emotions), respectively.

Line 1129ff: This section is really very speculative. The connection with the experience of birth is quite improbable.

Line 1186ff: One way in which the theory is not feminist at all is in that it starts with the mother schema, and then claims that attachment to God is equivalent to the infant's attachment to the mother. In that case, wouldn't monotheistic religions imagine their god as female? I only know God as a bearded old man.

The subject of this paper is timely. There has been considerable work about the mother’s infant schema (the “cute schema”, but since Konrad Lorenz’ early work on imprinting of goslings there has been virtually no interest in the converse, the infant’s mother schema. The main importance, and that could be stated more clearly in this paper, is that both the mother schema and the mother’s infant schema are the source of adult attachment and love. Presumably, the same brain mechanisms that are used for the mother-infant relationship are also used for adult relationships, as has been shown in the case of oxytocin which obviously acquired its role in social bonding initially in the form of triggering maternal behavior. Also worth mentioning is that all mammals are supposed to have both the mother’s infant schema and the infant’s mother schema, but that only social species have co-opted the brain mechanisms for social bonding among adults and only a few including the monogamous ones) are using them for bonding between mates.

There is also the question of the conceptual separation between the schemas, which are quite specific sensory-cognitive structures dependent on sensory input, and the emotional “resources” that are activated by the schemas. In drug-induced ecstasy, for example, or in music, emotions are triggered that are normally triggered by the mechanisms of social bonding. We can say that these emotional resources exist for the purpose of being triggered by mother schema and related schemata, but the emotion as such is different from the schema. Awe, for example, can be triggered by the mother schema, but also by interactions with a powerful alpha male. Presumably, the same emotional circuits but different triggers.

Some specific points:

Line 32: I wouldn't identify Cromagnon with Homo sapiens sapiens. The latter is the branch of H. sapiens that evolved in Africa at the time of the Neanderthals and Denisovans, and finally took over the world. The term Cromagnon should be limited to the earliest modern human race in Europe.

Line 54: Perhaps behavioral modernity is not such a great mystery at all, if there has been selection for higher intelligence for a long time. Intelligence would be considered a general-purpose mechanism that allows abstract concepts, including "mental time travel" to imagine future states of the world and the consequences of one's present and intended actions. Being of general usefulness, intelligence is responsible for a whole bunch of behaviors that archaeologists describe as “behavioral modernity”, as well as for those that historians describe as “civilization”.

Line 70-71: We are not really dealing here with "fetal cognitive abilities", but with cognitive abilities that start developing in the fetus but that reach a fully functional state of maturity only when the child starts talking. Linguistic ability is based on simple associative learning: a predisposition to associate sound patterns with things that are perceived through the senses, and the ability and motivation to reproduce these perceived sound patterns through the vocal apparatus. Later on, associations are not only with perceived things but also with abstract concepts that are in turn associated with "real" things that can be perceived through the senses.

Line 73: Here you will have to explain why a “prenatal mother schema” should exist in the first place. I cannot see any evolutionary advantage in having it. After birth it is important for the infant to recognize the mother and behave in a way that encourages continued maternal care and prevents infanticide. However, there is no such selective pressure before birth.

Line 85ff: Here you seem to assume that the mother schema develops in the fetus and infant and then persists as such in the adult, buried somehow in the depths of the mind but sometimes triggered by religious/ritualistic kinds of stimuli. Perhaps a more productive way of looking at it is that during ontogenesis the mother schema is transformed into something else, something more general. It seems to be obvious that adult forms of love encompass caring (derived both ontogenetically and phylogenetically from maternal care) and attachment (derived in large part from the infant's mother schema. In a parallel strand of ontogenetic development, the infant's more general experience of all-powerful adults develops into the perception of power hierarchies. It should be obvious that these two parallel ontogenetic paths, from mother schema to adult attachments and from the experience of infant dependence on powerful adults to dominance hierarchies, are the foundations of religious experience. And, it should be emphasized, the entirety of human sociality.

Line 91-92: Why should there be prenatal episodic memory? It makes no evolutionary sense. Memory is for learning, and learning makes sense only when it can influence future behaviour in adaptive ways. The fetus cannot manipulate the mother's behaviour through cognitive means, and things learned during fetal life would be useless after birth when the survival challenges are entirely different. If there is no survival advantage to prenatal memory, either it didn’t evolve at all or if it did, only as a byproduct of something that is indeed adaptive. The latter could only be postnatal episodic memory, but humans are born so prematurely that it is even doubtful that infants in the first year after birth have much of it.

Line 101ff: Somewhere you will have to explain what exactly "reflective consciousness" is, and what it isn't.

Line 130-132: Again, I doubt that things learned before birth can in any way be useful after birth. Also, being born at a more immature stage, as is the case in humans compared to other primates, implies lower, not higher ability to learn during the first months after birth.

Line 166ff: Suggestion: For religion, you may consider a conceptual distinction between two sources of religion. The first, which is directly derived from mother schema and social dominance, is the idea of very powerful sentient beings (gods) that may or may not interfere with human affairs. The second is derived from the contemplation of man's place in the world and especially his mortality, meaning it requires a higher stage of cognitive development. This second element produced the concept of mind-body dualism, as in the brahman-atman equivalence of the Vedanta and ideas of an afterlife and/or reincarnation. There may be different evolutionary, neurological and ontogenetic sources of these two kinds of "religious" ideas.

Line 224-225: Obviously, "complex reflective language" did not emerge suddenly. It must have evolved from a protolanguage that was based on nothing more than the ability, and the inclination, to associate complex sound patterns with things in the real world. This ability and inclination could evolve only in the presence of an evolutionary advantage. This evolutionary advantage was given only when human ancestors had acquired the ability to produce these complex sounds. It has been speculated that anatomical changes in the vocal apparatus associated with upright walking enabled speech. If so, the evolution of a protolanguage may date back several million years and may have driven brain evolution from Australopithecus to early Homo.

Line 230-231: The human-chimpanzee split happened perhaps a bit earlier, because mutation rates are lower than originally assumed, and therefore the evolutionary clock had to be recalibrated.

Line 323-314: Absolutely dispensable, or absolutely indispensable?

Line 346ff: Here, it seems, the argument is made that the mother-infant relationship is the source of selective pressure for the evolution of (proto)language. Let's not forget that language develops only at the age of about 2 years, while mortality is highest in the first year after birth. Motherese looks more like a hybrid of language and emotional expression, used at a time when baby’s linguistic development is still rudimentary, although it could be argued that protolanguage (though not more complex, grammatical language) evolved in part for mother-infant interactions and teaching.

Line 391-392: Music and religion arose in the context of ritual? This would suggest that ritual came before religion and music. As far as I know, ritual is a relative latecomer in cultural evolution, being strongly related to cultural complexity. See Zern, D. S. (1984). Religiousness related to cultural complexity and pressures to obey cultural norms. Genetic psychology monographs 110: 207-227. So, more likely religion and music were there for some time already when people started with rituals.

Line 458ff: Most of these postures have not much to do with the fetal position but are part of the repertoire of behaviors related to social dominance: making oneself smaller and defenseless by bowing, kneeling, prostration etc. These are gestures of submission to an alpha male, not re-enactments of the fetal position. They are used in interactions with a personal god, whereas a more upright position in meditation is related not to appeasement of a personal god but to an approach to the unknowable mysteries of life and existence. You may be more on target with dance, because being carried by the mother has a strong soothing effect on the infant, and such preference for rhythmic movement may persist into adulthood when it is expressed as dance.

Line 469ff: Here, I think, you should better work out the ontogenetic changes. There is infant attachment to the mother. In adult life, people depend no longer on the mother but they do depend on the groups to which they belong. This certainly was the case for prehistoric hunter-gatherer bands. What plausibly happens is that the neurological systems for infant attachment to the mother are developed into a new function: supporting the individual’s attachment to family, country, or any other group with which he identifies. Mother attachment as the evolutionary and ontogenetic source of positive (but not negative) ethnocentrism.

Line 541ff: Here we are dealing less with an activation of the mother schema as such, but with an activation of the pool of emotional resources that ordinarily become activated by the mother schema in infants and by related stimuli (friends, family members, or more abstract group identities) in adulthood.

Line 847-848: Transnatal memory is predicted to be not robust at all because experiences before birth are unlikely to lead to appropriate responses after birth, which are, during the first months at least, almost purely reflexive anyhow. Again, memory is learning, and learning makes sense only when the conditions at learning are maintained long enough for appropriate behavioural responses at a later time. When the conditions change as radically as they do at birth, any prenatal learning will be useless after birth.

Line 909: Oxytocin is an interesting case because empirical studies show its involvement in adult bonding although its primary function is in parturition and lactation. It shows that the evolutionary and ontogenetic source of adult bonding lies in the mother-infant relationship, since one of the original functions of oxytocin appears to be to activate the mother's care for the infant after birth. You may point out that so far we know of no biochemical marker for the mother schema, no chemical we know of that specifically triggers infant attachment to the mother, but it is perfectly plausible that the infant's mother schema carries over into adult life the way the mother's infant schema carries over into adult life, that indeed the two merge during ontogenesis to form stable adult attachments. Dopamine and opioids appear to be associated with general emotional resources that can be described as positive-euphoriant and tranquilizing (repressing negative emotions), respectively.

Line 1129ff: This section is really very speculative. The connection with the experience of birth is quite improbable.

Line 1186ff: One way in which the theory is not feminist at all is in that it starts with the mother schema, and then claims that attachment to God is equivalent to the infant's attachment to the mother. In that case, wouldn't monotheistic religions imagine their god as female? I only know God as a bearded old man.

Author Response

The subject of this paper is timely. There has been considerable work about the mother’s infant schema (the “cute schema”, but since Konrad Lorenz’ early work on imprinting of goslings there has been virtually no interest in the converse, the infant’s mother schema.

** I inserted: “Since Lorenz’ early work on imprinting of goslings there has been surprisingly interest in this multimodal cognitive representation of the mother from the fetal or infant perspective.”

The main importance, and that could be stated more clearly in this paper, is that both the mother schema and the mother’s infant schema are the source of adult attachment and love. Presumably, the same brain mechanisms that are used for the mother-infant relationship are also used for adult relationships, as has been shown in the case of oxytocin which obviously acquired its role in social bonding initially in the form of triggering maternal behavior. Also worth mentioning is that all mammals are supposed to have both the mother’s infant schema and the infant’s mother schema, but that only social species have co-opted the brain mechanisms for social bonding among adults and only a few including the monogamous ones) are using them for bonding between mates.

** I have now addressed all these points at various points. For example, I inserted the following: Taken together, the two schemas may be considered the foundation of carer-infant attachment and love. // If similar physiological mechanisms (e.g., oxytocin) enable and motivate adult relationships, the same two schemas may underlie all loving human relationships, including altruistic behaviors that “maximize inclusive fitness through the care of helpless offspring” (Preston, 2013, p. 1305). Both altruistic responding and offspring care require “(a) participation by nonmothers, (b) motor competence and expertise, (c) an adaptive opponency between avoidance and approach, and a facilitating role of (d) neonatal vulnerability, (e) salient distress, and (f) rewarding close contact. Physiologically, they also share neurohormonal support from (g) oxytocin, (h) the domain-general mesolimbocortical system, (i) the cingulate cortex, and (j) the orbitofrontal cortex. (Preston, 2013, p. 1305).“ // Oxytocin and vasopressin are also involved in the monogamous behaviors (affiliation, pair bonding) of non-human animals such as prairie voles (Young et al., 2001). Similar principles apply to non-human primates, other mammals, and other eusocial species (Maestripieri & Zehr, 1998): “Even species that are highly divergent from mammals, such as squid, crocodiles, clownfish, and rattlesnakes demonstrate functionally similar behaviors to sequester and protect young from predators during their most vulnerable developmental stage, shortly after birth.” (Preston, 2013, p. 1314, citing Hrdy, 2009).

There is also the question of the conceptual separation between the schemas, which are quite specific sensory-cognitive structures dependent on sensory input, and the emotional “resources” that are activated by the schemas. In drug-induced ecstasy, for example, or in music, emotions are triggered that are normally triggered by the mechanisms of social bonding. We can say that these emotional resources exist for the purpose of being triggered by mother schema and related schemata, but the emotion as such is different from the schema. Awe, for example, can be triggered by the mother schema, but also by interactions with a powerful alpha male. Presumably, the same emotional circuits but different triggers.

** I inserted: “There is considerable overlap between emotions evoked by MS, the infant schema, and other schemas or stimuli, suggesting these schemas might be related to each other. Emotions typical of drug-induced ecstasy, music, and social bonding are also similar to each other. Awe, for example, can be triggered by MS or interactions with a powerful alpha male.”

Some specific points:

Line 32: I wouldn't identify Cromagnon with Homo sapiens sapiens. The latter is the branch of H. sapiens that evolved in Africa at the time of the Neanderthals and Denisovans, and finally took over the world. The term Cromagnon should be limited to the earliest modern human race in Europe.

** The reference to Cromagnon has been deleted.

Line 54: Perhaps behavioral modernity is not such a great mystery at all, if there has been selection for higher intelligence for a long time. Intelligence would be considered a general-purpose mechanism that allows abstract concepts, including "mental time travel" to imagine future states of the world and the consequences of one's present and intended actions. Being of general usefulness, intelligence is responsible for a whole bunch of behaviors that archaeologists describe as “behavioral modernity”, as well as for those that historians describe as “civilization”.

** I added this: “At first glance, it (behavioral modernity) might be explicable simply by selection for higher intelligence. But it also involves universal, quasi-irrational behaviors such as religion and music. Intelligence alone can hardly imply or predict universal beliefs in anthropomorphized supernatural agents with whom humans communicate cognitively and emotionally. Nor can it imply or predict the universal existence of musical rituals involving transcendental experiences.”

Line 70-71: We are not really dealing here with "fetal cognitive abilities", but with cognitive abilities that start developing in the fetus but that reach a fully functional state of maturity only when the child starts talking. Linguistic ability is based on simple associative learning: a predisposition to associate sound patterns with things that are perceived through the senses, and the ability and motivation to reproduce these perceived sound patterns through the vocal apparatus. Later on, associations are not only with perceived things but also with abstract concepts that are in turn associated with "real" things that can be perceived through the senses.

** To justify the term “fetal cognitive abilities” I added the following just before first use: “Phenotypic variation in the cognitive abilities of newborns meant that the more cognitively able were more likely to survive to reproductive age. Whereas this principle presumably applies to all mammals, the obstetric dilemma in early humans meant that infant survival depended increasingly directly on the active contribution of infants to maternal-infant bonding.”

**I also inserted this: “Motherese is not language, which emerges about two years later. But the ontogenetic emergence of language relies on perceptual, cognitive, motor, and social foundations that were laid in infancy and even before birth (May et al., 2011).

** I also explained multimodal associative learning, e.g. “Non-human primates communicate by means of complex multimodal combinations of gestural, vocal, and facial signals (Slocombe et al., 2011). Primate infants are predisposed to learn such patterns and their meanings in trial-and-error processes of playful experimentation, usually in social-interactive settings. Early humans presumably did the same, but with a larger vocabulary and more complex grammar”

Line 73: Here you will have to explain why a “prenatal mother schema” should exist in the first place. I cannot see any evolutionary advantage in having it. After birth it is important for the infant to recognize the mother and behave in a way that encourages continued maternal care and prevents infanticide. However, there is no such selective pressure before birth.

** The following has been inserted: “Phenotypic variation in the cognitive abilities of newborns meant that the more cognitively able were more likely to survive to reproductive age. Whereas this principle presumably applies to all mammals, the obstetric dilemma in early humans meant that infant survival depended increasingly directly on the active contribution of infants to maternal-infant bonding.”

Line 85ff: Here you seem to assume that the mother schema develops in the fetus and infant and then persists as such in the adult, buried somehow in the depths of the mind but sometimes triggered by religious/ritualistic kinds of stimuli. Perhaps a more productive way of looking at it is that during ontogenesis the mother schema is transformed into something else, something more general. It seems to be obvious that adult forms of love encompass caring (derived both ontogenetically and phylogenetically from maternal care) and attachment (derived in large part from the infant's mother schema. In a parallel strand of ontogenetic development, the infant's more general experience of all-powerful adults develops into the perception of power hierarchies. It should be obvious that these two parallel ontogenetic paths, from mother schema to adult attachments and from the experience of infant dependence on powerful adults to dominance hierarchies, are the foundations of religious experience. And, it should be emphasized, the entirety of human sociality.

** The following has been added: “The feelings of caring and attachment that emerged when MS was evoked motivated the development of religious power/dominance hierarchies. These parallel ontogenetic-psychological paths—from mother to adult attachment and from powerful adults to dominance hierarchies—may represent the psychological foundations of religious experience and other aspects of human sociality.”

** I also change this sentence: “MS (or a generalized ontogenetic transformation of MS) was activated in children and adults in ritual-like situations that were somehow similar to the prenatal situation.”

Line 91-92: Why should there be prenatal episodic memory? It makes no evolutionary sense. Memory is for learning, and learning makes sense only when it can influence future behaviour in adaptive ways. The fetus cannot manipulate the mother's behaviour through cognitive means, and things learned during fetal life would be useless after birth when the survival challenges are entirely different. If there is no survival advantage to prenatal memory, either it didn’t evolve at all or if it did, only as a byproduct of something that is indeed adaptive. The latter could only be postnatal episodic memory, but humans are born so prematurely that it is even doubtful that infants in the first year after birth have much of it.

** There is now a new section on transnatal memory in which its nature is clarified: “Transnatal memory is procedural (implicit), not episodic; the infant does not “remember” birth or any preceding or following event until it acquires simple language.”

** I also added “In the absence of prenatal/infant episodic memory (Mullally & Maguire, 2014)…”

Line 101ff: Somewhere you will have to explain what exactly "reflective consciousness" is, and what it isn't.

** It is now defined as “the overt ability to observe one’s own consciousness and exercise introspection”

Line 130-132: Again, I doubt that things learned before birth can in any way be useful after birth.

** I inserted a new footnote: “The word “learn” is used here in the usual psychological sense that applies to human and non-human animals equally. Learning means acquiring new information, habits, or abilities. Any contact with the environment that causes a lasting behavioral change (including physical responses or perceptual/cognitive changes) may be considered learning (cf. Schwartz, 1989).” The usefulness of prenatally acquired “knowledge” for mother-infant attachment is empirically demonstrated by for example Mastropieri and Turkewitz (1999).

Also, being born at a more immature stage, as is the case in humans compared to other primates, implies lower, not higher ability to learn during the first months after birth.

** I believe this has now been more clearly explained. There was selective pressure to learn faster at an earlier ontogenetic stage.

Line 166ff: Suggestion: For religion, you may consider a conceptual distinction between two sources of religion. The first, which is directly derived from mother schema and social dominance, is the idea of very powerful sentient beings (gods) that may or may not interfere with human affairs. The second is derived from the contemplation of man's place in the world and especially his mortality, meaning it requires a higher stage of cognitive development. This second element produced the concept of mind-body dualism, as in the brahman-atman equivalence of the Vedanta and ideas of an afterlife and/or reincarnation. There may be different evolutionary, neurological and ontogenetic sources of these two kinds of "religious" ideas.

** These points have been addressed at different points, as follows.

“A clear answer should distinguish between social/emotional and cognitive/philosophical aspects.”

“These parallel ontogenetic-psychological paths—from mother to adult attachment and from powerful adults to dominance hierarchies—may represent the psychological foundations of religious experience and other aspects of human sociality.”

“The religions of the world offer diverse answers to two central questions: Where did the universe come from? What happens to the “soul” or “spirit” after death? The prehistoric emergence of language and reflective consciousness made such questions possible. We might suppose that both questions were posed by all ancient peoples. In pre-scientific culture, it seemed obvious that the soul left the dead body and continued to exist in some form. To explain the origin of the universe, it is not surprising that ancient peoples came up with creative agents (creators) and creation myths.”

Line 224-225: Obviously, "complex reflective language" did not emerge suddenly. It must have evolved from a protolanguage that was based on nothing more than the ability, and the inclination, to associate complex sound patterns with things in the real world. This ability and inclination could evolve only in the presence of an evolutionary advantage. This evolutionary advantage was given only when human ancestors had acquired the ability to produce these complex sounds. It has been speculated that anatomical changes in the vocal apparatus associated with upright walking enabled speech. If so, the evolution of a protolanguage may date back several million years and may have driven brain evolution from Australopithecus to early Homo.

** The gradual nature of these changes has been emphasized at several points. I added this sentence: “Examples include anatomical changes in the vocal apparatus associated with upright walking (Provine, 2004).”

Line 230-231: The human-chimpanzee split happened perhaps a bit earlier, because mutation rates are lower than originally assumed, and therefore the evolutionary clock had to be recalibrated.

** I replaced “about” by “at least”.

Line 323-314: Absolutely dispensable, or absolutely indispensable?

** Indispensable, corrected

Line 346ff: Here, it seems, the argument is made that the mother-infant relationship is the source of selective pressure for the evolution of (proto)language. Let's not forget that language develops only at the age of about 2 years, while mortality is highest in the first year after birth. Motherese looks more like a hybrid of language and emotional expression, used at a time when baby’s linguistic development is still rudimentary, although it could be argued that protolanguage (though not more complex, grammatical language) evolved in part for mother-infant interactions and teaching.

** The following was inserted: “Motherese is not language, which emerges about two years later. But the ontogenetic emergence of language relies on perceptual, cognitive, motor, and social foundations that were laid in infancy and even before birth (May et al., 2011).”

Line 391-392: Music and religion arose in the context of ritual? This would suggest that ritual came before religion and music. As far as I know, ritual is a relative latecomer in cultural evolution, being strongly related to cultural complexity. See Zern, D. S. (1984). Religiousness related to cultural complexity and pressures to obey cultural norms. Genetic psychology monographs 110: 207-227. So, more likely religion and music were there for some time already when people started with rituals.

** I have now written “Ritual is not confined to humans. A general definition includes “those stylized displays reported by ethologists to occur among the birds, the beast and even the insects” (Rappaport, 1999, p. 25). Eilam et al. (2006) considered that “motor rituals are characterized by their close linkage to a few environmental locations and the repeated performance of relatively few acts” (p. 456).”

Line 458ff: Most of these postures have not much to do with the fetal position but are part of the repertoire of behaviors related to social dominance: making oneself smaller and defenseless by bowing, kneeling, prostration etc. These are gestures of submission to an alpha male -not e-enactments of the fetal position. They are used in interactions with a personal god, whereas a more upright position in meditation is related not to appeasement of a personal god but to an approach to the unknowable mysteries of life and existence. You may be more on target with dance, because being carried by the mother has a strong soothing effect on the infant, and such preference for rhythmic movement may persist into adulthood when it is expressed as dance.

** I inserted this: “Alternatively, ritual postures can be explained by the power difference between a worshipper and her/his god, and the desire to depict and feel humility (e.g., Hamdan, 2010). In social dominance hierarchies, individuals make themselves smaller and more vulnerable by adopting closed or bent body positions (bowing, kneeling, prostration), for example when submitting to an alpha male or to a dominant female or male (Cashdan, 1998; Hawley et al., 2008). In meditation, a more upright position is adopted because appeasement of a personal god is not necessary or because bent postures cannot be maintained for long periods.”

Line 469ff: Here, I think, you should better work out the ontogenetic changes. There is infant attachment to the mother. In adult life, people depend no longer on the mother but they do depend on the groups to which they belong. This certainly was the case for prehistoric hunter-gatherer bands. What plausibly happens is that the neurological systems for infant attachment to the mother are developed into a new function: supporting the individual’s attachment to family, country, or any other group with which he identifies. Mother attachment as the evolutionary and ontogenetic source of positive (but not negative) ethnocentrism.

** I inserted this: “Adult ritual participants depend on each other just as infants depend on their mothers, but to a lesser degree. The link between the two cases may be an example of Piaget’s (1953, 1975, 2006) assimilation and accommodation (cf. Bowlby, 1969). Neurological and physiological systems for infant-mother attachment may be transformed or adapted to support the individual’s attachment to—and psychosocial identification with—family, clan, country, or other group.”

Line 541ff: Here we are dealing less with an activation of the mother schema as such, but with an activation of the pool of emotional resources that ordinarily become activated by the mother schema in infants and by related stimuli (friends, family members, or more abstract group identities) in adulthood.

** I added the following. “Instead, they explained their strong emotions and changed states in terms of adult relationships and situations: families, other groups and social identities, and musical/religious rituals.”

Line 847-848: Transnatal memory is predicted to be not robust at all because experiences before birth are unlikely to lead to appropriate responses after birth, which are, during the first months at least, almost purely reflexive anyhow. Again, memory is learning, and learning makes sense only when the conditions at learning are maintained long enough for appropriate behavioural responses at a later time. When the conditions change as radically as they do at birth, any prenatal learning will be useless after birth.

** I clarified that this is about procedural (implicit) memory, i.e. not episodic. I have also clarified the usefulness of prenatal learning for postnatal attachment, and explained how prenatal MS is transformed into postnatal MS at birth, an example of accommodation.

Line 909: Oxytocin is an interesting case because empirical studies show its involvement in adult bonding although its primary function is in parturition and lactation. It shows that the evolutionary and ontogenetic source of adult bonding lies in the mother-infant relationship, since one of the original functions of oxytocin appears to be to activate the mother's care for the infant after birth. You may point out that so far we know of no biochemical marker for the mother schema, no chemical we know of that specifically triggers infant attachment to the mother, but it is perfectly plausible that the infant's mother schema carries over into adult life the way the mother's infant schema carries over into adult life, that indeed the two merge during ontogenesis to form stable adult attachments. Dopamine and opioids appear to be associated with general emotional resources that can be described as positive-euphoriant and tranquilizing (repressing negative emotions), respectively.

** Oytocin is referred to several times. I clarified the transition from oxytocin-motivated postnatal behavior and later MS-motivated behavior “In these ways, human infants actively promote infant-carer bonding in the longer term—beyond the more immediate effects of birth hormones such as oxytocin.”) I added “Dopamine is linked to more positive-euphoriant emotions, whereas opioids are more tranquilizing.” I hesitated to write “No known biochemical specifically triggers infant attachment to the mother” because I could not find a literature source for this claim.

Line 1129ff: This section is really very speculative. The connection with the experience of birth is quite improbable.

** The entire section on the “light of god” and the accompanying references have been deleted (although the other reviewer liked it).

Line 1186ff: One way in which the theory is not feminist at all is in that it starts with the mother schema, and then claims that attachment to God is equivalent to the infant's attachment to the mother. In that case, wouldn't monotheistic religions imagine their god as female? I only know God as a bearded old man.

**A new section has been added. “5.6. Male Dominance. If gods and spirits are cultural transformations of the mother as perceived by the fetus and infant, why are gods are usually male? First, social structures are usually patriarchal; a male god is one of many consequences of male dominance (Suchocki, 1994). Second, the fetus and infant have no concept of gender. // Infants do prefer high-pitched speech and singing (Trainor & Zacharias, 1998), faces that adults consider attractive (Ramsey et al., 2004), and adults that smile and maintain eye contact (Blass & Camp, 2001). They are sensitive to the size of strangers (children versus large/small adults), upon which fear responses depend (Brooks & Lewis, 1976). Gender differences in infants (under one year of age) have been demonstrated in visual interest for toys (Alexander et al., 2009) and people versus machines (Connellan et al., 2000). // But there is no evidence that infants have a concept of gender or understand gender distinctions, even though parents or other adults may treat baby girls and boys differently (Trehub et al., 1997). Verbal gender distinctions in infants emerge at around 21 months (Zosuls et al. 2009), after which girls start to exhibit greater empathy than boys (Hastings et al., 2000).”

Round 2

Reviewer 2 Report

There are only a few more points that you should consider, most of them rather minor:

There are only a few more points that you should consider, most of them rather minor:

Lines 44-45:

Awkward wording. The two are of course distinct because one happens in the genome and the other consists of human behaviour. Unless the cited source conceptualizes it differently, the better phrasing would be that the two interact. The concept of gene-culture coevolution means that culture influences the selective pressures that act on the genes (e.g., milk drinking selecting for lactose tolerance, or infanticide and contraception selecting for different behavioural dispositions), and that genetic predispositions influence the kinds of culture that people create (e.g., higher "genotypic intelligence" favouring greater cultural complexity).

Lines 57-62:

You can be more specific than this. What the listed examples demonstrate is the ability to take elements of emotion and cognition out of the context for which they evolved, for example, feelings of attachment and deference that evolved for the mother-child relationship and for social dominance hierarchies are taken out of their natural context to create religion. You can point out that this ability is not part of what we call intelligence. Actually, of course, the everyday use of intelligence does require a presumably evolved ability to take bits and pieces of knowledge about the world and put them together to create novel conclusions. Perhaps humans are a species in which this ability tends to generalize to intra-psychic processes, such that bits and pieces of instinct and emotion are plastered together to form religions and other non-scientific, ideological worldviews.

Line 93 paragraph:

Here you proceed straight from the infant's mother schema to the expressions in art, religion etc. Perhaps we should have here a paragraph explaining that there were certain advantages in carrying the psychological adaptations for MS over into adult life, and that this is the reason why MS doesn't stop abruptly when children are old enough to be on their own. You can hypothesize that these psychological structures were useful for maintaining group cohesion at a time when this was necessary for survival, and that by infusing them into the psychological adaptations for social dominance hierarchies, these hierarchies were transformed into the kind of leadership structures that we encounter today in functioning (but not dysfunctional) businesses and governments.

Line 117:

Dominance hierarchies more likely existed independent of the MS, based on aggression alone, like the pecking order in the chicken yard. I would rather assume that the nature of dominance hierarchies got transformed (a little bit at least) when MS was infused into them (mainly for subordinates) together with caring (mainly for leaders).

Lines 155-156:

The nature of this question makes it impossible to determine whether reflective consciousness is unique to humans and whether it is more complex in humans than in other animals, unless we can specify a behavioural output that cannot exist without it.

Line 175 paragraph:

This is where the changes in the vocal apparatus belong that you mention in line 172-173, because this allowed complex vocal output. Without this, more complex forms of speech could not have evolved. What seems to have happened is that after the evolution of bipedalism and some restructuring of the larynx, the vocal apparatus became capable of producing a greater variety of sounds. This made it possible for words to evolve, and once there were words (perhaps starting more than 2 million years ago), the benefits from vocal communication were so large that there was selective pressure for brain enlargement and cognitive evolution. Interestingly, any language-like communication system that uses discrete words to express not only feelings but to inform about things and events in the environment has to be analytical in the sense that it deals with discrete entities that, in later stages of cognitive-linguistic evolution, are placed in hierarchical relationships to create not only grammar, but a cognitive style that is more analytic and less holistic than cognition in other mammals. You can say, the right hemisphere is chimpanzee-like but the left hemisphere represents the kind of thinking typical for humans.

Line 536:

Just as a comment: You seem to assume that Motherese is ancient enough to be a precursor of music. It is often believed that mother-child interactions became more playful and intense only in recent times when women started having fewer children and more leisure. I would think that mothers in simple hunter-gatherer societies have too much else to do and little time to interact very extensively with their infants, beyond the necessities. I am wondering if there is any ethnographic evidence about motherese in hunter-gatherers.

Line 1020:

Distress reduced by morphine and naloxone? Naloxone is a morphine antagonist, so we would expect opposite effects. More likely, distress is reduced by morphine and the morphine effect is abolished by naloxone. If distress is enhanced by naloxone alone, it would mean that most likely endorphins (acting on mu receptors) are reducing distress in the unmedicated state.

Line 1255:

Theories are “socially constructed”? To me this term implies that they are bogus, essentially lies that the protagonist is pushing for some ulterior reason. At least, this is how "socially constructed” is understood in the sciences, as an expression of epistemic relativism.

Line 1288:

"Early gender-specific behaviors"? Don't feminists insist that there are no gender-specific behaviors, and such ideas are the product of male chauvinists? Better change this, or else the feminists will get you fired!

Lines 1295-1296:

Fairness is not well explained by carer-infant interactions. It is better explained by reciprocal exchange (Trivers' "reciprocal altruism"). Fairness, like reciprocity and commercial transactions, implies interactions between equals, which is not what the infant-carer relationship is. A more plausible thing that is explained by the carer-infant relationship is that women are more religious than men. But this is again an anti-feminist dogma because it assumes that women are more adapted for this kind of relationship than are men.

There are only a few more points that you should consider, most of them rather minor:

Lines 44-45:

Awkward wording. The two are of course distinct because one happens in the genome and the other consists of human behaviour. Unless the cited source conceptualizes it differently, the better phrasing would be that the two interact. The concept of gene-culture coevolution means that culture influences the selective pressures that act on the genes (e.g., milk drinking selecting for lactose tolerance, or infanticide and contraception selecting for different behavioural dispositions), and that genetic predispositions influence the kinds of culture that people create (e.g., higher "genotypic intelligence" favouring greater cultural complexity).

Lines 57-62:

You can be more specific than this. What the listed examples demonstrate is the ability to take elements of emotion and cognition out of the context for which they evolved, for example, feelings of attachment and deference that evolved for the mother-child relationship and for social dominance hierarchies are taken out of their natural context to create religion. You can point out that this ability is not part of what we call intelligence. Actually, of course, the everyday use of intelligence does require a presumably evolved ability to take bits and pieces of knowledge about the world and put them together to create novel conclusions. Perhaps humans are a species in which this ability tends to generalize to intra-psychic processes, such that bits and pieces of instinct and emotion are plastered together to form religions and other non-scientific, ideological worldviews.

Line 93 paragraph:

Here you proceed straight from the infant's mother schema to the expressions in art, religion etc. Perhaps we should have here a paragraph explaining that there were certain advantages in carrying the psychological adaptations for MS over into adult life, and that this is the reason why MS doesn't stop abruptly when children are old enough to be on their own. You can hypothesize that these psychological structures were useful for maintaining group cohesion at a time when this was necessary for survival, and that by infusing them into the psychological adaptations for social dominance hierarchies, these hierarchies were transformed into the kind of leadership structures that we encounter today in functioning (but not dysfunctional) businesses and governments.

Line 117:

Dominance hierarchies more likely existed independent of the MS, based on aggression alone, like the pecking order in the chicken yard. I would rather assume that the nature of dominance hierarchies got transformed (a little bit at least) when MS was infused into them (mainly for subordinates) together with caring (mainly for leaders).

Lines 155-156:

The nature of this question makes it impossible to determine whether reflective consciousness is unique to humans and whether it is more complex in humans than in other animals, unless we can specify a behavioural output that cannot exist without it.

Line 175 paragraph:

This is where the changes in the vocal apparatus belong that you mention in line 172-173, because this allowed complex vocal output. Without this, more complex forms of speech could not have evolved. What seems to have happened is that after the evolution of bipedalism and some restructuring of the larynx, the vocal apparatus became capable of producing a greater variety of sounds. This made it possible for words to evolve, and once there were words (perhaps starting more than 2 million years ago), the benefits from vocal communication were so large that there was selective pressure for brain enlargement and cognitive evolution. Interestingly, any language-like communication system that uses discrete words to express not only feelings but to inform about things and events in the environment has to be analytical in the sense that it deals with discrete entities that, in later stages of cognitive-linguistic evolution, are placed in hierarchical relationships to create not only grammar, but a cognitive style that is more analytic and less holistic than cognition in other mammals. You can say, the right hemisphere is chimpanzee-like but the left hemisphere represents the kind of thinking typical for humans.

Line 536:

Just as a comment: You seem to assume that Motherese is ancient enough to be a precursor of music. It is often believed that mother-child interactions became more playful and intense only in recent times when women started having fewer children and more leisure. I would think that mothers in simple hunter-gatherer societies have too much else to do and little time to interact very extensively with their infants, beyond the necessities. I am wondering if there is any ethnographic evidence about motherese in hunter-gatherers.

Line 1020:

Distress reduced by morphine and naloxone? Naloxone is a morphine antagonist, so we would expect opposite effects. More likely, distress is reduced by morphine and the morphine effect is abolished by naloxone. If distress is enhanced by naloxone alone, it would mean that most likely endorphins (acting on mu receptors) are reducing distress in the unmedicated state.

Line 1255:

Theories are “socially constructed”? To me this term implies that they are bogus, essentially lies that the protagonist is pushing for some ulterior reason. At least, this is how "socially constructed” is understood in the sciences, as an expression of epistemic relativism.

Line 1288:

"Early gender-specific behaviors"? Don't feminists insist that there are no gender-specific behaviors, and such ideas are the product of male chauvinists? Better change this, or else the feminists will get you fired!

Lines 1295-1296:

Fairness is not well explained by carer-infant interactions. It is better explained by reciprocal exchange (Trivers' "reciprocal altruism"). Fairness, like reciprocity and commercial transactions, implies interactions between equals, which is not what the infant-carer relationship is. A more plausible thing that is explained by the carer-infant relationship is that women are more religious than men. But this is again an anti-feminist dogma because it assumes that women are more adapted for this kind of relationship than are men.

There are only a few more points that you should consider, most of them rather minor:

Lines 44-45:

Awkward wording. The two are of course distinct because one happens in the genome and the other consists of human behaviour. Unless the cited source conceptualizes it differently, the better phrasing would be that the two interact. The concept of gene-culture coevolution means that culture influences the selective pressures that act on the genes (e.g., milk drinking selecting for lactose tolerance, or infanticide and contraception selecting for different behavioural dispositions), and that genetic predispositions influence the kinds of culture that people create (e.g., higher "genotypic intelligence" favouring greater cultural complexity).

Lines 57-62:

You can be more specific than this. What the listed examples demonstrate is the ability to take elements of emotion and cognition out of the context for which they evolved, for example, feelings of attachment and deference that evolved for the mother-child relationship and for social dominance hierarchies are taken out of their natural context to create religion. You can point out that this ability is not part of what we call intelligence. Actually, of course, the everyday use of intelligence does require a presumably evolved ability to take bits and pieces of knowledge about the world and put them together to create novel conclusions. Perhaps humans are a species in which this ability tends to generalize to intra-psychic processes, such that bits and pieces of instinct and emotion are plastered together to form religions and other non-scientific, ideological worldviews.

Line 93 paragraph:

Here you proceed straight from the infant's mother schema to the expressions in art, religion etc. Perhaps we should have here a paragraph explaining that there were certain advantages in carrying the psychological adaptations for MS over into adult life, and that this is the reason why MS doesn't stop abruptly when children are old enough to be on their own. You can hypothesize that these psychological structures were useful for maintaining group cohesion at a time when this was necessary for survival, and that by infusing them into the psychological adaptations for social dominance hierarchies, these hierarchies were transformed into the kind of leadership structures that we encounter today in functioning (but not dysfunctional) businesses and governments.

Line 117:

Dominance hierarchies more likely existed independent of the MS, based on aggression alone, like the pecking order in the chicken yard. I would rather assume that the nature of dominance hierarchies got transformed (a little bit at least) when MS was infused into them (mainly for subordinates) together with caring (mainly for leaders).

Lines 155-156:

The nature of this question makes it impossible to determine whether reflective consciousness is unique to humans and whether it is more complex in humans than in other animals, unless we can specify a behavioural output that cannot exist without it.

Line 175 paragraph:

This is where the changes in the vocal apparatus belong that you mention in line 172-173, because this allowed complex vocal output. Without this, more complex forms of speech could not have evolved. What seems to have happened is that after the evolution of bipedalism and some restructuring of the larynx, the vocal apparatus became capable of producing a greater variety of sounds. This made it possible for words to evolve, and once there were words (perhaps starting more than 2 million years ago), the benefits from vocal communication were so large that there was selective pressure for brain enlargement and cognitive evolution. Interestingly, any language-like communication system that uses discrete words to express not only feelings but to inform about things and events in the environment has to be analytical in the sense that it deals with discrete entities that, in later stages of cognitive-linguistic evolution, are placed in hierarchical relationships to create not only grammar, but a cognitive style that is more analytic and less holistic than cognition in other mammals. You can say, the right hemisphere is chimpanzee-like but the left hemisphere represents the kind of thinking typical for humans.

Line 536:

Just as a comment: You seem to assume that Motherese is ancient enough to be a precursor of music. It is often believed that mother-child interactions became more playful and intense only in recent times when women started having fewer children and more leisure. I would think that mothers in simple hunter-gatherer societies have too much else to do and little time to interact very extensively with their infants, beyond the necessities. I am wondering if there is any ethnographic evidence about motherese in hunter-gatherers.

Line 1020:

Distress reduced by morphine and naloxone? Naloxone is a morphine antagonist, so we would expect opposite effects. More likely, distress is reduced by morphine and the morphine effect is abolished by naloxone. If distress is enhanced by naloxone alone, it would mean that most likely endorphins (acting on mu receptors) are reducing distress in the unmedicated state.

Line 1255:

Theories are “socially constructed”? To me this term implies that they are bogus, essentially lies that the protagonist is pushing for some ulterior reason. At least, this is how "socially constructed” is understood in the sciences, as an expression of epistemic relativism.

Line 1288:

"Early gender-specific behaviors"? Don't feminists insist that there are no gender-specific behaviors, and such ideas are the product of male chauvinists? Better change this, or else the feminists will get you fired!

Lines 1295-1296:

Fairness is not well explained by carer-infant interactions. It is better explained by reciprocal exchange (Trivers' "reciprocal altruism"). Fairness, like reciprocity and commercial transactions, implies interactions between equals, which is not what the infant-carer relationship is. A more plausible thing that is explained by the carer-infant relationship is that women are more religious than men. But this is again an anti-feminist dogma because it assumes that women are more adapted for this kind of relationship than are men.

There are only a few more points that you should consider, most of them rather minor:

Lines 44-45:

Awkward wording. The two are of course distinct because one happens in the genome and the other consists of human behaviour. Unless the cited source conceptualizes it differently, the better phrasing would be that the two interact. The concept of gene-culture coevolution means that culture influences the selective pressures that act on the genes (e.g., milk drinking selecting for lactose tolerance, or infanticide and contraception selecting for different behavioural dispositions), and that genetic predispositions influence the kinds of culture that people create (e.g., higher "genotypic intelligence" favouring greater cultural complexity).

Lines 57-62:

You can be more specific than this. What the listed examples demonstrate is the ability to take elements of emotion and cognition out of the context for which they evolved, for example, feelings of attachment and deference that evolved for the mother-child relationship and for social dominance hierarchies are taken out of their natural context to create religion. You can point out that this ability is not part of what we call intelligence. Actually, of course, the everyday use of intelligence does require a presumably evolved ability to take bits and pieces of knowledge about the world and put them together to create novel conclusions. Perhaps humans are a species in which this ability tends to generalize to intra-psychic processes, such that bits and pieces of instinct and emotion are plastered together to form religions and other non-scientific, ideological worldviews.

Line 93 paragraph:

Here you proceed straight from the infant's mother schema to the expressions in art, religion etc. Perhaps we should have here a paragraph explaining that there were certain advantages in carrying the psychological adaptations for MS over into adult life, and that this is the reason why MS doesn't stop abruptly when children are old enough to be on their own. You can hypothesize that these psychological structures were useful for maintaining group cohesion at a time when this was necessary for survival, and that by infusing them into the psychological adaptations for social dominance hierarchies, these hierarchies were transformed into the kind of leadership structures that we encounter today in functioning (but not dysfunctional) businesses and governments.

Line 117:

Dominance hierarchies more likely existed independent of the MS, based on aggression alone, like the pecking order in the chicken yard. I would rather assume that the nature of dominance hierarchies got transformed (a little bit at least) when MS was infused into them (mainly for subordinates) together with caring (mainly for leaders).

Lines 155-156:

The nature of this question makes it impossible to determine whether reflective consciousness is unique to humans and whether it is more complex in humans than in other animals, unless we can specify a behavioural output that cannot exist without it.

Line 175 paragraph:

This is where the changes in the vocal apparatus belong that you mention in line 172-173, because this allowed complex vocal output. Without this, more complex forms of speech could not have evolved. What seems to have happened is that after the evolution of bipedalism and some restructuring of the larynx, the vocal apparatus became capable of producing a greater variety of sounds. This made it possible for words to evolve, and once there were words (perhaps starting more than 2 million years ago), the benefits from vocal communication were so large that there was selective pressure for brain enlargement and cognitive evolution. Interestingly, any language-like communication system that uses discrete words to express not only feelings but to inform about things and events in the environment has to be analytical in the sense that it deals with discrete entities that, in later stages of cognitive-linguistic evolution, are placed in hierarchical relationships to create not only grammar, but a cognitive style that is more analytic and less holistic than cognition in other mammals. You can say, the right hemisphere is chimpanzee-like but the left hemisphere represents the kind of thinking typical for humans.

Line 536:

Just as a comment: You seem to assume that Motherese is ancient enough to be a precursor of music. It is often believed that mother-child interactions became more playful and intense only in recent times when women started having fewer children and more leisure. I would think that mothers in simple hunter-gatherer societies have too much else to do and little time to interact very extensively with their infants, beyond the necessities. I am wondering if there is any ethnographic evidence about motherese in hunter-gatherers.

Line 1020:

Distress reduced by morphine and naloxone? Naloxone is a morphine antagonist, so we would expect opposite effects. More likely, distress is reduced by morphine and the morphine effect is abolished by naloxone. If distress is enhanced by naloxone alone, it would mean that most likely endorphins (acting on mu receptors) are reducing distress in the unmedicated state.

Line 1255:

Theories are “socially constructed”? To me this term implies that they are bogus, essentially lies that the protagonist is pushing for some ulterior reason. At least, this is how "socially constructed” is understood in the sciences, as an expression of epistemic relativism.

Line 1288:

"Early gender-specific behaviors"? Don't feminists insist that there are no gender-specific behaviors, and such ideas are the product of male chauvinists? Better change this, or else the feminists will get you fired!

Lines 1295-1296:

Fairness is not well explained by carer-infant interactions. It is better explained by reciprocal exchange (Trivers' "reciprocal altruism"). Fairness, like reciprocity and commercial transactions, implies interactions between equals, which is not what the infant-carer relationship is. A more plausible thing that is explained by the carer-infant relationship is that women are more religious than men. But this is again an anti-feminist dogma because it assumes that women are more adapted for this kind of relationship than are men.

There are only a few more points that you should consider, most of them rather minor:

Lines 44-45:

Awkward wording. The two are of course distinct because one happens in the genome and the other consists of human behaviour. Unless the cited source conceptualizes it differently, the better phrasing would be that the two interact. The concept of gene-culture coevolution means that culture influences the selective pressures that act on the genes (e.g., milk drinking selecting for lactose tolerance, or infanticide and contraception selecting for different behavioural dispositions), and that genetic predispositions influence the kinds of culture that people create (e.g., higher "genotypic intelligence" favouring greater cultural complexity).

Lines 57-62:

You can be more specific than this. What the listed examples demonstrate is the ability to take elements of emotion and cognition out of the context for which they evolved, for example, feelings of attachment and deference that evolved for the mother-child relationship and for social dominance hierarchies are taken out of their natural context to create religion. You can point out that this ability is not part of what we call intelligence. Actually, of course, the everyday use of intelligence does require a presumably evolved ability to take bits and pieces of knowledge about the world and put them together to create novel conclusions. Perhaps humans are a species in which this ability tends to generalize to intra-psychic processes, such that bits and pieces of instinct and emotion are plastered together to form religions and other non-scientific, ideological worldviews.

Line 93 paragraph:

Here you proceed straight from the infant's mother schema to the expressions in art, religion etc. Perhaps we should have here a paragraph explaining that there were certain advantages in carrying the psychological adaptations for MS over into adult life, and that this is the reason why MS doesn't stop abruptly when children are old enough to be on their own. You can hypothesize that these psychological structures were useful for maintaining group cohesion at a time when this was necessary for survival, and that by infusing them into the psychological adaptations for social dominance hierarchies, these hierarchies were transformed into the kind of leadership structures that we encounter today in functioning (but not dysfunctional) businesses and governments.

Line 117:

Dominance hierarchies more likely existed independent of the MS, based on aggression alone, like the pecking order in the chicken yard. I would rather assume that the nature of dominance hierarchies got transformed (a little bit at least) when MS was infused into them (mainly for subordinates) together with caring (mainly for leaders).

Lines 155-156:

The nature of this question makes it impossible to determine whether reflective consciousness is unique to humans and whether it is more complex in humans than in other animals, unless we can specify a behavioural output that cannot exist without it.

Line 175 paragraph:

This is where the changes in the vocal apparatus belong that you mention in line 172-173, because this allowed complex vocal output. Without this, more complex forms of speech could not have evolved. What seems to have happened is that after the evolution of bipedalism and some restructuring of the larynx, the vocal apparatus became capable of producing a greater variety of sounds. This made it possible for words to evolve, and once there were words (perhaps starting more than 2 million years ago), the benefits from vocal communication were so large that there was selective pressure for brain enlargement and cognitive evolution. Interestingly, any language-like communication system that uses discrete words to express not only feelings but to inform about things and events in the environment has to be analytical in the sense that it deals with discrete entities that, in later stages of cognitive-linguistic evolution, are placed in hierarchical relationships to create not only grammar, but a cognitive style that is more analytic and less holistic than cognition in other mammals. You can say, the right hemisphere is chimpanzee-like but the left hemisphere represents the kind of thinking typical for humans.

Line 536:

Just as a comment: You seem to assume that Motherese is ancient enough to be a precursor of music. It is often believed that mother-child interactions became more playful and intense only in recent times when women started having fewer children and more leisure. I would think that mothers in simple hunter-gatherer societies have too much else to do and little time to interact very extensively with their infants, beyond the necessities. I am wondering if there is any ethnographic evidence about motherese in hunter-gatherers.

Line 1020:

Distress reduced by morphine and naloxone? Naloxone is a morphine antagonist, so we would expect opposite effects. More likely, distress is reduced by morphine and the morphine effect is abolished by naloxone. If distress is enhanced by naloxone alone, it would mean that most likely endorphins (acting on mu receptors) are reducing distress in the unmedicated state.

Line 1255:

Theories are “socially constructed”? To me this term implies that they are bogus, essentially lies that the protagonist is pushing for some ulterior reason. At least, this is how "socially constructed” is understood in the sciences, as an expression of epistemic relativism.

Line 1288:

"Early gender-specific behaviors"? Don't feminists insist that there are no gender-specific behaviors, and such ideas are the product of male chauvinists? Better change this, or else the feminists will get you fired!

Lines 1295-1296:

Fairness is not well explained by carer-infant interactions. It is better explained by reciprocal exchange (Trivers' "reciprocal altruism"). Fairness, like reciprocity and commercial transactions, implies interactions between equals, which is not what the infant-carer relationship is. A more plausible thing that is explained by the carer-infant relationship is that women are more religious than men. But this is again an anti-feminist dogma because it assumes that women are more adapted for this kind of relationship than are men.

Author Response

There are only a few more points that you should consider, most of them rather minor:

Lines 44-45 (“The distinction between biological and cultural evolution, or between nature and nurture, is not necessarily distinct (Tomasello & Slobin, 2004).”)

Awkward wording. The two are of course distinct because one happens in the genome and the other consists of human behaviour. Unless the cited source conceptualizes it differently, the better phrasing would be that the two interact. The concept of gene-culture coevolution means that culture influences the selective pressures that act on the genes (e.g., milk drinking selecting for lactose tolerance, or infanticide and contraception selecting for different behavioural dispositions), and that genetic predispositions influence the kinds of culture that people create (e.g., higher "genotypic intelligence" favouring greater cultural complexity).

** Thank you. I think at this stage I shorten at this point, so the new sentence reads “Biological and cultural evolution interact (Tomasello & Slobin, 2004).”

Lines 57-62 (“At first glance, it might be explicable simply by selection for higher intelligence. But it also involves universal, quasi-irrational behaviors such as religion and music. Intelligence alone can hardly imply or predict universal beliefs in anthropomorphized supernatural agents with whom humans communicate cognitively and emotionally. Nor can it imply or predict the universal existence of musical rituals involving transcendental experiences.”)

You can be more specific than this. What the listed examples demonstrate is the ability to take elements of emotion and cognition out of the context for which they evolved, for example, feelings of attachment and deference that evolved for the mother-child relationship and for social dominance hierarchies are taken out of their natural context to create religion. You can point out that this ability is not part of what we call intelligence. Actually, of course, the everyday use of intelligence does require a presumably evolved ability to take bits and pieces of knowledge about the world and put them together to create novel conclusions. Perhaps humans are a species in which this ability tends to generalize to intra-psychic processes, such that bits and pieces of instinct and emotion are plastered together to form religions and other non-scientific, ideological worldviews.

 ** The new passage: “At first glance, it might be explicable simply by selection for higher intelligence. But behavioral modernity also involves universal, quasi-irrational behaviors such as religion and music. While these could involve taking elements of emotion and cognition out of the context for which they evolved (for example, infant attachment) and “intelligently” applying them elsewhere, beliefs in anthropomorphized supernatural agents with whom humans communicate cognitively and emotionally contradict accepted elements of intelligence such as logic, reasoning, critical thinking, and problem solving. The universality of religious beliefs and musical rituals involving transcendental experiences suggest the existence of an additional factor.”

Line 93 paragraph (“Operant conditioning. Animal behavior can be conditioned by rewards and punishments. As MS became more salient and complex, it was more often activated later in life (during childhood and adulthood). In the context of early human ritual, repetitive stimulus patterns similar to those perceived before birth or during infancy evoked MS and associated emotions. These mysterious feelings (and corresponding biochemicals such as opioids and neuropeptides) acted as rewards (positive reinforcements), motivating ritual participants to repeat the actions that appeared to have caused the feelings. In proto-religious rituals, MS was evoked by combinations of subdued light, low-frequency acoustic resonances, unusual smells or tastes, unusual body postures, and/or changed states of consciousness brought about by diverse means, such that participants sensed the presence of mysterious supernatural agents. For music, early forms of singing, drumming, or dance evoked MS if they were perceived as similar to maternal voice, heartbeat/footsteps, or body movements during walking from the fetal perspective.”)

Here you proceed straight from the infant's mother schema to the expressions in art, religion etc. Perhaps we should have here a paragraph explaining that there were certain advantages in carrying the psychological adaptations for MS over into adult life, and that this is the reason why MS doesn't stop abruptly when children are old enough to be on their own. You can hypothesize that these psychological structures were useful for maintaining group cohesion at a time when this was necessary for survival, and that by infusing them into the psychological adaptations for social dominance hierarchies, these hierarchies were transformed into the kind of leadership structures that we encounter today in functioning (but not dysfunctional) businesses and governments.

** This point has been addressed elsewhere and the passage in question has been revised as follows. “Rates of infant mortality in ancient societies were high, consistent with the salience and complexity of mother and infant schemas and their strong biochemical foundation. On that basis, we might expect MS to remain functional long after infancy, even if it no longer contributes directly to survival or reproduction. MS-activation may strengthen and thereby benefit other relationships, and hence individual and group survival and reproduction. If so, MS in children and adults is not vestigial, like the human appendix, but instead contributes positively to all human relationships throughout the lifespan.”

Line 117 (“These parallel ontogenetic-psychological paths—from mother to adult attachment and from powerful adults to dominance hierarchies—may represent the psychological foundations of religious experience and other aspects of human sociality.”)

Dominance hierarchies more likely existed independent of the MS, based on aggression alone, like the pecking order in the chicken yard. I would rather assume that the nature of dominance hierarchies got transformed (a little bit at least) when MS was infused into them (mainly for subordinates) together with caring (mainly for leaders).

** Ok. The new text reads: “These parallel ontogenetic-psychological paths—from MS to adult attachment and from powerful adults to dominance hierarchies—may have reinforced the psychological foundations of religious experience and other aspects of human sociality, strengthening existing dominance hierarchies that were previously based on aggression alone.”

Lines 155-156 (“Why is human reflective consciousness unique—or if it is not unique, why is it so complex by comparison to other forms of consciousness in non-human animals?”)

The nature of this question makes it impossible to determine whether reflective consciousness is unique to humans and whether it is more complex in humans than in other animals, unless we can specify a behavioural output that cannot exist without it.

** The new sentence reads: “Why is human reflective consciousness, as explored empirically and developmentally by Zelazo (2004), so complex by comparison to consciousness in non-human animals?” (The more fundamental problem as discussed by Nagel and others is addressed elsewhere.)

Line 175 paragraph (“Language. How and why did human communication become so enormously complex in its arbitrary sound patterns, extensive vocabularies, and hierarchical structures—relative to the poor linguistic abilities of non-human primates? The literature addressing this question is old, extensive, diverse, and unresolved.”)

This is where the changes in the vocal apparatus belong that you mention in line 172-173, because this allowed complex vocal output. Without this, more complex forms of speech could not have evolved. What seems to have happened is that after the evolution of bipedalism and some restructuring of the larynx, the vocal apparatus became capable of producing a greater variety of sounds. This made it possible for words to evolve, and once there were words (perhaps starting more than 2 million years ago), the benefits from vocal communication were so large that there was selective pressure for brain enlargement and cognitive evolution. Interestingly, any language-like communication system that uses discrete words to express not only feelings but to inform about things and events in the environment has to be analytical in the sense that it deals with discrete entities that, in later stages of cognitive-linguistic evolution, are placed in hierarchical relationships to create not only grammar, but a cognitive style that is more analytic and less holistic than cognition in other mammals. You can say, the right hemisphere is chimpanzee-like but the left hemisphere represents the kind of thinking typical for humans.

** I added the following. “Whereas a descended larynx may initially have had nothing to do with language (Fitch & Reby, 2001), postnatal laryngeal descent (Nishimura, 2003) and increased laryngeal control (Hickok, 2017) clearly played important roles in language evolution, increasing the number of distinguishable sounds (Esling, 2012). Brain enlargement may have led to laryngeal descent (Ciani & Chiarelli, 1992) and vice-versa (Nishimura et al., 2006): the benefits of more complex communication may have meant selection for encephalization (Dunbar, 1993). What ultimately triggered or drove these complex interacting changes is unclear.”

** I guess that psycholinguistic questions of grammar, cognitive style, and hemispheric specialization, while certainly relevant, are too complex for the present text, or beyond its scope. Either that or the changes would be too big at this stage.

Line 536 (“Motherese is an interesting candidate for the origin of music because it similarly involves emotional contagion and mutual empathy (Hatfield et al., 2011): monitoring the emotional state of the other, joint attention, and imitation (Papoušek, 1996).”)

Just as a comment: You seem to assume that Motherese is ancient enough to be a precursor of music. It is often believed that mother-child interactions became more playful and intense only in recent times when women started having fewer children and more leisure. I would think that mothers in simple hunter-gatherer societies have too much else to do and little time to interact very extensively with their infants, beyond the necessities. I am wondering if there is any ethnographic evidence about motherese in hunter-gatherers.

** Here is the new text. “Motherese in modern industrialized countries can hardly explain the origin of music if it was absent in ancient hunter-gatherer societies. Studies of modern hunter-gatherer cultures may be relevant here. Hewlett and Roulette (2016) reported that “natural pedagogy was impacted by the Aka cultural context and interactions relied more on touch, physical proximity and pointing, and less on verbal exchange and motherese” (p. 10). But in an embodied cognition approach (Yu et al., 2005), touch (physical proximity) and gesture (e.g., pointing) are as important for motherese as vocalizations, and inseparable from them. Ancient proto-motherese presumably also involved emotional contagion and mutual empathy (Hatfield et al., 2011): monitoring the emotional state of the other, joint attention, and imitation (Papoušek, 1996).”

Line 1020 (“Herman and Panksepp (1978) showed that symptoms of separation-induced distress in guinea pigs can be reduced by morphine and naloxone.”)

Distress reduced by morphine and naloxone? Naloxone is a morphine antagonist, so we would expect opposite effects. More likely, distress is reduced by morphine and the morphine effect is abolished by naloxone. If distress is enhanced by naloxone alone, it would mean that most likely endorphins (acting on mu receptors) are reducing distress in the unmedicated state.

** Thanks for picking that up. I guess the paper does not need this level of detail, so the sentence and reference to Herman and Panksepp have been deleted.

Line 1255 („Theories of the origin of religion and music are socially constructed (Clayton, 2013) and co-1255 determined by the author’s social, historic, cultural and political context (cf. Chenail, 2011).”)

Theories are “socially constructed”? To me this term implies that they are bogus, essentially lies that the protagonist is pushing for some ulterior reason. At least, this is how "socially constructed” is understood in the sciences, as an expression of epistemic relativism.

** The term “socially constructed” and the reference to Clayton (2013) have been deleted.

Line 1288 (“The present theory is feminist in the sense that modern human behavior is held to be based 1287 primarily on early gender-specific female behaviors. In this way, the theory responds to accusations 1288 of sexism in evolutionary theory (Fedigan, 1986).”)

"Early gender-specific behaviors"? Don't feminists insist that there are no gender-specific behaviors, and such ideas are the product of male chauvinists? Better change this, or else the feminists will get you fired!

** To clarify, I added “in connection with childcare in ancient hunter-gatherer societies”

Lines 1295-1296 (“Today’s world religions are dominated by male leaders, but women have always dominated 1295 early moral training (Lamb, 2014). Carer-infant interactions can parsimoniously explain…“)

Fairness is not well explained by carer-infant interactions. It is better explained by reciprocal exchange (Trivers' "reciprocal altruism"). Fairness, like reciprocity and commercial transactions, implies interactions between equals, which is not what the infant-carer relationship is. A more plausible thing that is explained by the carer-infant relationship is that women are more religious than men. But this is again an anti-feminist dogma because it assumes that women are more adapted for this kind of relationship than are men.

** Here is the new text. “Whereas carers and infants are not equals, which makes carer-infant interactions a poor model for adult interactions, carer-infant relationships are nevertheless founded on mutual empathy (Decety et al., 2016)—an ability that is crucial for both carers and infants if the infant is to survive in dangerous situations. In this way, carer-infant interactions can parsimoniously explain aspects of…”